# α-/γ-Taxilin are required for centriolar subdistal appendage assembly and microtubule organization

**Dandan Ma[1], Fulin Wang[1], Rongyi Wang[1], Yingchun Hu[2], Zhiquan Chen[1], Ning Huang[1], Yonglu Tian[1], Yuqing Xia[1], Junlin Teng[2]\*, Jianguo Chen[1,3]\***

[1]Key Laboratory of Cell Proliferation and Differentiation of the Ministry of Education, College of Life Sciences, Peking University, Beijing, China; [2]Core Facilities College of Life Sciences, Peking University, Beijing, China; [3]Center for Quantitative Biology, Peking University, Beijing, China

**Abstract** The centrosome composed of a pair of centrioles (mother and daughter) and pericentriolar material, and is mainly responsible for microtubule nucleation and anchorage in animal cells. The subdistal appendage (SDA) is a centriolar structure located at the mother centriole's subdistal region, and it functions in microtubule anchorage. However, the molecular composition and detailed structure of the SDA remain largely unknown. Here, we identified α-taxilin and γ-taxilin as new SDA components that form a complex via their coiled-coil domains and that serve as a new subgroup during SDA hierarchical assembly. The taxilins' SDA localization is dependent on ODF2, and α-taxilin recruits CEP170 to the SDA. Functional analyses suggest that α- and γ-taxilin are responsible for SDA structural integrity and centrosomal microtubule anchorage during interphase and for proper spindle orientation during metaphase. Our results shed light on the molecular components and functional understanding of the SDA hierarchical assembly and microtubule organization.

## Editor's evaluation

The subdistal appendage (SDA) is a distinct structure of the mother centriole that functions in anchoring microtubules, orientation of spindles and formation of the basal foot of cilia. Though many components in this structure have been identified, two new proteins, i.e. α- and γ-taxilins are identified in this work. By using super-resolution microscopy, biochemical and genetic tools, the precise localization and function of these new proteins have been defined, which would deepen our understanding of this unique structure.

**\*For correspondence:**
junlinteng@pku.edu.cn (JT);
chenjg@pku.edu.cn (JC)

## Introduction

The centrosome, the main microtubule organizing center (MTOC) in many eukaryotic cells, participates in microtubule-related activities that include cell division (*Cabral et al., 2019*; *Wu et al., 2012*), cell polarity maintenance (*Burute et al., 2017*), cell signaling transduction (*Barvitenko et al., 2018*), and ciliogenesis (*Pitaval et al., 2017*; *Tu et al., 2018*). The centrosome is a non-membrane-bound organelle composed of a pair of orthogonally arranged centrioles and surrounded by pericentriolar material (PCM) (*Delattre and Gönczy, 2004*). The two centrioles, mother and daughter, are distinguished from each other by the decorations at the distal and subdistal ends of the mother centriole, called the distal/subdistal appendages (DAs/SDAs) (*Tischer et al., 2021*).

Accumulated data have revealed the structural characteristics and functions of DAs and SDAs. DA proteins (i.e. CEP83, CEP89, FBF1, SCLT1, and CEP164) are essential for centriole-to-membrane

docking and TTBK2 recruitment that promote ciliogenesis (*Tanos et al., 2013*; *Huang et al., 2018*). Super-resolution microscopy has enabled researchers to identify the precise localization of centrosomal components, including those of the DA and SDA (*Yang et al., 2018*), and it has revealed that SDAs differ from DAs in both molecular composition and intracellular functions (*Uzbekov and Alieva, 2018*). Located beneath the DAs, SDAs are composed of ODF2, CEP128, centriolin, NdeI, CCDC68, CCDC120, ninein, and CEP170, etc. (*Huang et al., 2017*; *Kashihara et al., 2019*), and they are classified into two groups according to their specific localizations. The ODF group (ODF2, CEP128, and centriolin) is localized on SDAs only, whereas the ninein group (ninein, Kif2a, p150glued, CCDC68, CCDC120, and CEP170) are localized on both SDAs and the proximal ends of centrioles (*Mazo et al., 2016*; *Huang et al., 2017*). Of them, ODF2 occupies the bases of both the DAs and SDAs via different domains and thus participates in the hierarchical assembly of both DAs and SDAs (*Tateishi et al., 2013*; *Chong et al., 2020*). Also, ODF2 ensures that microtubules focus properly at the centrosome, which then controls microtubule organization and stability (*Hung et al., 2016a*), and centriolin, ninein, CCDC68, and CCDC120 are all required for microtubule stabilization and anchorage at the centrosomes (*Gromley et al., 2003*; *Mogensen et al., 2000*; *Huang et al., 2017*). The SDA proteins cooperate to form a microtubule anchoring complex that maintains the microtubule array. Although this complex has been well described, most likely other unidentified SDA proteins still exist.

The taxilin family (i.e. α-taxilin, β-taxilin, and γ-taxilin) have been reported to interact with syntaxin family members and to participate in intracellular vesicle transport (*Nogami et al., 2003*). Here, we have identified α- and γ-taxilin as new SDA structural components. After they are recruited to SDA via ODF2, they form ring-like structures between CCDC120 and ninein, and participate in CEP170 assembly. Like other colleagues in the SDAs, α- and γ-taxilin are involved in microtubule anchorage at the centrosome and are also indispensable for spindle orientation during metaphase.

## Results

### Screening of α-taxilin and γ-taxilin as new SDA components

APEX2-mediated proximal labeling is an approach that uses hydrogen peroxide ($H_2O_2$) as an oxidant to catalyze biotin-phenol (BP), a small molecular substrate, to produce the reactive BP radical that conjugates to the endogenous proteins that are proximal to APEX2 (*Hung et al., 2016b*). To search for new SDA components, we labeled two previously identified SDA components CCDC68 and CCDC120 (*Huang et al., 2017*) by infusing them with V5-tagged APEX2. When overexpressed in HeLa cells, the V5 immunostained CCDC68 and CCDC120 fusion proteins showed a ring (top view) (*Figure 1—figure supplement 1A*) or three dots (side view) (*Figure 1—figure supplement 1B*) at the centrosome, results that were consistent with their SDA locations when they had been stained by their specific antibodies (*Huang et al., 2017*). After treatment with BP and $H_2O_2$, the biotinylated endogenous proteins co-localized and encircled V5-tagged CCDC120 or V5-tagged CCDC68 (*Figure 1—figure supplement 1A-B*), thus indicating that APEX2-mediated biotinylation had successfully marked the endogenous proteins around CCDC68 or CCDC120 at the centrosomes. As a negative control, samples without BP or $H_2O_2$ showed no biotinylation signal. Furthermore, immunoblotting results also confirmed the endogenous biotinylated proteins mediated by CCDC68 and CCDC120 proximal labeling (*Figure 1—figure supplement 1C-D*). Finally, the biotinylated proteins were enriched by Streptavidin-coated beads and sent for mass spectrographic (MS) analysis to search for CCDC68 and CCDC120 proximal candidates.

Among those candidates for CCDC68 and CCDC120 proximal labeling, a series of previously identified centriolar proteins (marked in green in *Figure 1—figure supplement 1E*) were found, and thus verified the efficiency of our centrosomal proteomics approach. Since several centrosomal proteins possess the coiled-coil domain that is the basis for the protein-protein interactions essential for centrosome configuration (*Andersen et al., 2003*), we focused on proteins containing one or more coiled-coil domains. To determine their sub-cellular localizations, each protein candidate was tagged (e.g. V5, mNeonGreen, or pmEmerald) and expressed in U2OS cells. Other than the already known centrosome proteins, five new proteins, including α-taxilin, DACT1, DRG2, NCAPH2, and SMAP2 (*Figure 1—figure supplement 1F*), were found co-localized with centrosome markers CP110 or γ-tubulin, thus proving that they resided on centrosomes. Among them, DRG2 was found in both CCDC68 and CCDC120 proximal candidates; α-taxilin, NCAPH2, and SMAP2 were found in CCDC68 proximal

candidates; and DACT1 was found in CCDC120 proximal candidates (*Figure 1—figure supplement 1E-F*). A previous study had shown γ-taxilin co-localized with Nek2A at the centrosome during interphase (*Makiyama et al., 2018*). The immunofluorescence assay showed it co-localized with γ-tubulin (*Figure 1—figure supplement 1E*), thus confirming its centrosomal localization.

Among the proteins that showed centrosomal localization, α- and γ-taxilin are similar in that their middle regions each has a long coiled-coil domain (*Figure 1—figure supplement 2A*; *Makiyama et al., 2018*). Both the anti-α-taxilin antibody and anti-γ-taxilin antibody, which were designed to recognize the C-terminus of each of those proteins (*Figure 1—figure supplement 2A-C*), were chosen to detect their specificities. Immunoblotting results showed that the two proteins' molecular weights (above 72 kDa for α-taxilin and about 70 kDa for γ-taxilin) were slightly higher than predicted (*Figure 1—figure supplement 2B-C*). Immunoblotting also detected significant decreases in the amount of those proteins (*Figure 1—figure supplement 2B-C*). The immunofluorescence assay showed that α-taxilin and γ-taxilin stained by their antibodies focused prominently at the centrosome, as indicated by their co-localizations with γ-tubulin in human RPE-1 cells. Also, the fluorescence intensities of both taxilins decreased at the centrosome after siRNA knockdown (*Figure 1—figure supplement 2D*). Those results show both the antibodies' specialties and the effectiveness of siRNAs.

Detailed α- and γ-taxilin localization at the centrosome was determined using super-resolution microscopy with a 3D-structured illumination system (SIM). In the G1 phase, the top views of α- and γ-taxilin showed their ring-like patterns encompassing one centrin-3 dot that resides in the distal lumen of centrioles (*Middendorp et al., 1997*; *Figure 1A–B*). The α- and γ-taxilin side views showed three dots occupying two levels with one level (two dots) beside ODF2 and thus at the SDA (*Figure 1A–B*), and the other level (one dot) assumed to be at the proximal end (*Figure 1A–E*). Those patterns resembled those of the ninein group, such as ninein and CEP170 (*Mazo et al., 2016*). Indeed, α-taxilin clearly co-localized with the ninein and CEP170 rings in the top view, and with the three ninein and/or CEP170 dots in the side view (*Figure 1C and E*). Corresponding γ-taxilin patterns resembled those of α-taxilin by containing a smaller ring that resided inside the ninein and CEP170 signals (*Figure 1D and F*).

We continued by asking which segments of α-taxilin and γ-taxilin were responsible for their SDA and proximal localizations. To examine that, we expressed a series of deletion mutants in RPE-1 cells and assayed their immunofluorescence using CEP170 as a marker (*Figure 1G–H*, *Figure 1—figure supplement 2E-F*). The immunofluorescence assay showed that the M2 deletion (261–300 aa) in α-taxilin eliminated its localization at the SDA, and deletion of M3 (301–350 aa) and M4 (351–400 aa) eliminated its localization at the proximal end (*Figure 1I* and *Figure 1—figure supplement 1G*). Other deletion mutants (i.e. the N-terminus [1–185 aa], the C-terminus [492–546 aa] and M1 [186–260 aa]) did not affect α-taxilin's centrosomal localization patterns (*Figure 1—figure supplement 1G*). These data suggest that α-taxilin's M2 is responsible for its SDA localization, while M3 and M4 are required for its proximal end localization. For γ-taxilin, M1 (228–270 aa) is responsible for its localization at the proximal end, while M2 (271–317 aa) is responsible for its SDA localization (*Figure 1J* and *Figure 1—figure supplement 1H*).

## The precise SDA localizations of α-taxilin and γ-taxilin

Using stimulated emission depletion (STED) nanoscopy, we further characterized α- and γ-taxilin rings, comparing them with those of other SDA proteins. The STED images showed that the rings resembled those of the other SDA proteins, including ODF2, CCDC68, CCDC120, ninein, and CEP170 (*Figure 2A*). Of those, the ODF2 ring had the smallest diameter and the rest of those proteins' rings increased in size in this order: CCDC68, CCDC120, γ-taxilin, α-taxilin, ninein, and CEP170 (*Figure 2A–B*). Therefore, those SDA proteins formed ordered concentric circles beginning with ODF2 and ending with CEP170 in the following order: ODF2, CCDC68, CCDC120, γ-taxilin, α-taxilin, ninein, and CEP170.

We then measured SDA proteins' longitudinal positions by co-immunostaining with the proximal end protein C-Nap1 (*Vlijm et al., 2018*), which serves as a position reference (*Figure 2C–D*). ODF2 covered a wide range, occupying between about 360 nm and 589 nm distances from C-Nap1, and so it supposedly resides at the base of both the DA and SDA, respectively. This is comparable to the data obtained by a dSTORM super-resolution microscopy study (*Chong et al., 2020*). Expect for CCDC120, the rest of the SDA proteins had two layered positions with one layer proximal to C-Nap1 and the other localizing between the two ODF2 layers. The side view of CCDC120 showed

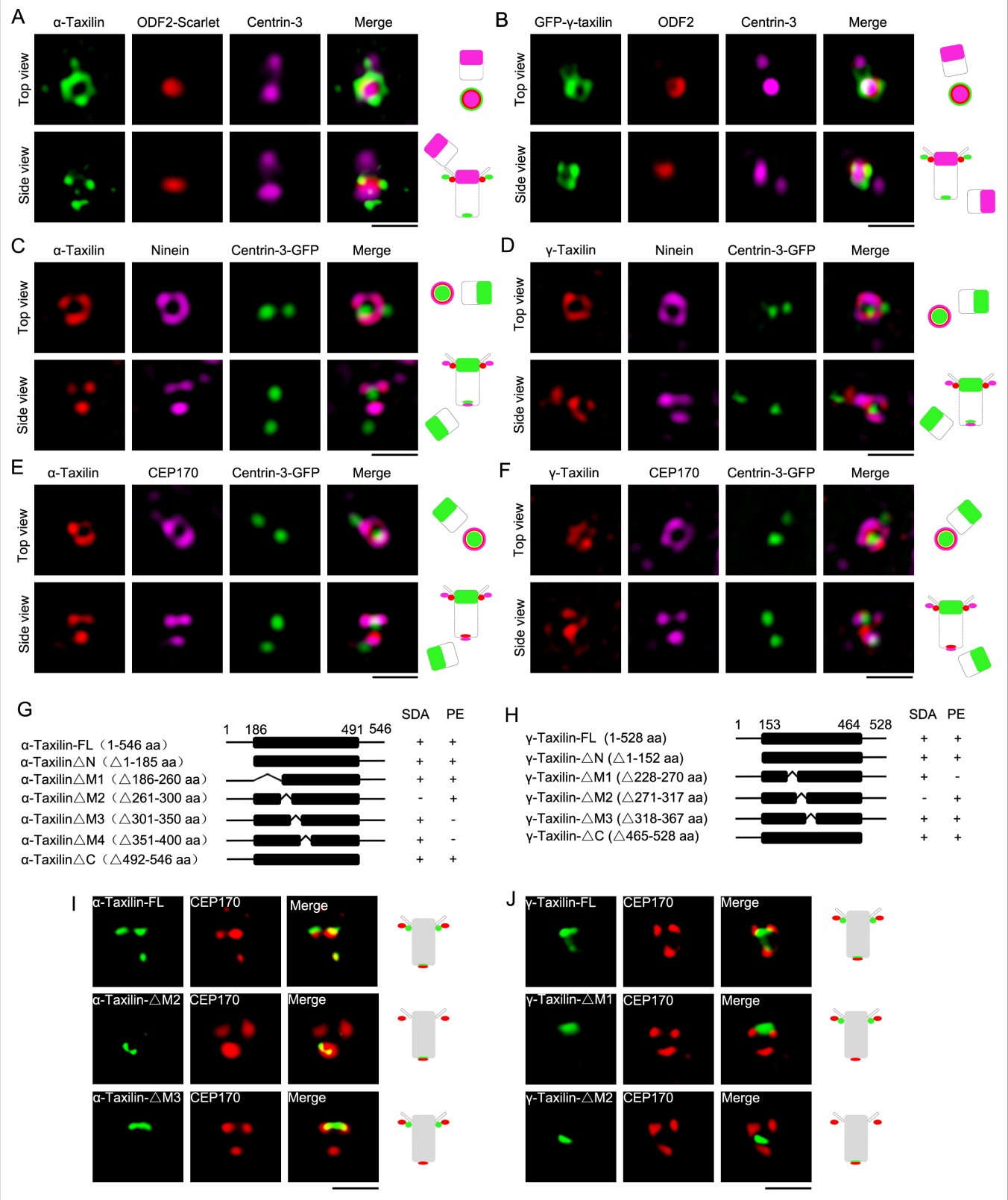

**Figure 1.** Structured illumination microscopy (SIM) images and characterizations of α-taxilin and γ-taxilin at the centrosomes. (**A**) Immunostained α-taxilin (green) and centrin-3 (magenta) in RPE-1 cells transfected with ODF2-Scarlet (red). Scale bar, 1 μm. The cartoons to the right of each set of images graphically depict the merge images. (**B**) Immunostained ODF2 (red) and centrin-3 (magenta) in RPE-1 cells transfected with GFP-γ-taxilin (green). Scale bar, 1 μm. (**C**) Immunostained α-taxilin (red) and ninein (magenta) in RPE-1 cells transfected with centrin-3-GFP (green). Scale bar, 1 μm.

*Figure 1 continued on next page*

*Figure 1 continued*

(**D**) Immunostained γ-taxilin (red) and ninein (magenta) in RPE-1 cells transfected with centrin-3-GFP (green). Scale bar, 1 µm. (**E**) Immunostained α-taxilin (red) and CEP170 (magenta) in RPE-1 cells transfected with centrin-3-GFP (green). Scale bar, 1 µm. (**F**) Immunostained γ-taxilin (red) and CEP170 (magenta) in RPE-1 cells transfected with centrin-3-GFP (green). Scale bar, 1 µm. (**G–H**) Schematic showing full-length (FL) and deletion mutants (Δ) of α-taxilin (**G**) and γ-taxilin (**H**). N, N terminus; M, middle; C, C terminus; +, positive; −, negative; SDA, subdistal appendages; PE, proximal end. (**I**) Immunofluorescence images of HA-tagged FL α-taxilin and deletion mutants (green) in (**G**) and CEP170 (red) in RPE-1 cells. Scale bar, 1 µm. (**J**) Immunofluorescence images of CEP170 (red) in RPE-1 cells transfected with GFP-tagged FL γ-taxilin and deletion mutants (green) in (**H**). Scale bar, 1 µm. Arrowheads in (**I**) and (**J**) show α-taxilin and γ-taxilin SDA localizations, respectively.

The online version of this article includes the following source data and figure supplement(s) for figure 1:

**Figure supplement 1.** α-Taxilin and γ-taxilin proposed as subdistal appendage proteins as shown by CCDC68 and CCDC120 proximal labeling.

**Figure supplement 1—source data 1.** Full immunoblots labeled and unlabeled for *Figure 1—figure supplement 1C and D*.

**Figure supplement 1—source data 2.** CCDC68 and CCDC120 proximity spectrometry data.

**Figure supplement 2.** α-Taxilin and γ-taxilin antibodies' specificities and localization characteristics.

**Figure supplement 2—source data 1.** Full immunoblots labeled and unlabeled for *Figure 1—figure supplement 2B, C, E and F*.

**Figure supplement 2—source data 2.** Data of normalized α-taxilin fluorescence intensity at centrosome of control- and α-taxilin-siRNA treated RPE-1 cells.

**Figure supplement 2—source data 3.** Data of normalized γ-taxilin fluorescence intensity at centrosome of control- and γ-taxilin-siRNA treated RPE-1 cells.

a three-layered pattern. At the proximal end, one layer resembled ninein group proteins; while the other two layers settled about 458 nm and 636 nm from C-Nap1 and were assumed to reside at the SDA and DA structures, respectively (*Figure 2C–D*).

By combining those SDA diameter and longitudinal position measurements seen in STED images, the relative localization of each SDA protein was established at the mother centriole. First, ODF2 resides closest to the centriole wall, while ninein and CEP170 reside at the tip of SDA. At the SDA structure, CCDC68 and CCDC120 localize close to ODF2, and α-taxilin and γ-taxilin are located in concentric circles between CCDC120 and ninein (*Figure 2B and E*). In the longitudinal position, γ-taxilin was higher than CCDC68, CCDC120, ninein and CEP170, while α-taxilin was lower than those proteins (*Figure 2D–E*).

## α-Taxilin and γ-taxilin localization at the SDA depends on ODF2

To reveal the assembly manners of α- and γ-taxilin in the SDAs, the interactions between them and other SDA proteins were characterized through immunoprecipitation. Since ODF2 is the innermost SDAs layer, it is the starting point for assembly of the other SDA components (e.g. TCHP, CCDC68, CCDC120, etc.) (*Ibi et al., 2011*; *Huang et al., 2017*). The immunoprecipitation results revealed that endogenous ODF2 interacted with both α- and γ-taxilin in RPE-1 cells (*Figure 3A–B*). Correspondingly, the specific localization of ODF2 and α- or γ-taxilin via STED nanoscopy is consistent with their specific localization as indicated in *Figure 2*. The top view of the SDA region showed that α- and γ-taxilin encircled the ODF2 ring (*Figure 3C–D*). When viewed from the side, α- and γ-taxilin localized proximal to ODF2's upper level (*Figure 3C–D*).

Furthermore, immunoblot analysis showed that when ODF2 was depleted by siRNA knockdown, α- and γ-taxilin levels in those cells did not change significantly (*Figure 3E–G*). However, fluorescence intensity of both α- and γ-taxilin decreased at the centrosomes after ODF2 depletion (*Figure 3—figure supplement 1A-B*), suggesting that ODF2 affects their centrosomal localizations without affecting their protein levels. As shown by 3D-SIM images, ODF2 depletion by siRNA treatment resulted in α-taxilin or γ-taxilin signal loss at the SDA region, while their proximal end signals were not affected (*Figure 3H–I*). Moreover, those lost SDA signals could be rescued by overexpressing full-length ODF2. However, the 1–59 aa deletion mutant, which affects ODF2 localization at the SDA (*Tateishi et al., 2013*) and recruitment of other SDA components (*Huang et al., 2017*), could not rescue SDA localization of either α-taxilin or γ-taxilin after siRNA treatment (*Figure 3H–I*). These data further illuminate the role of ODF2 in α- and γ-taxilin assembly at the SDAs.

In contrast, depleted CCDC68 or CCDC120 did not affect α-taxilin or γ-taxilin fluorescence intensity at centrosomes in RPE-1 cells (*Figure 3—figure supplement 1C-F*), and ectopically expressed CCDC68 and CCDC120 could not be immunoprecipitated from cell lysates by ectopically expressed

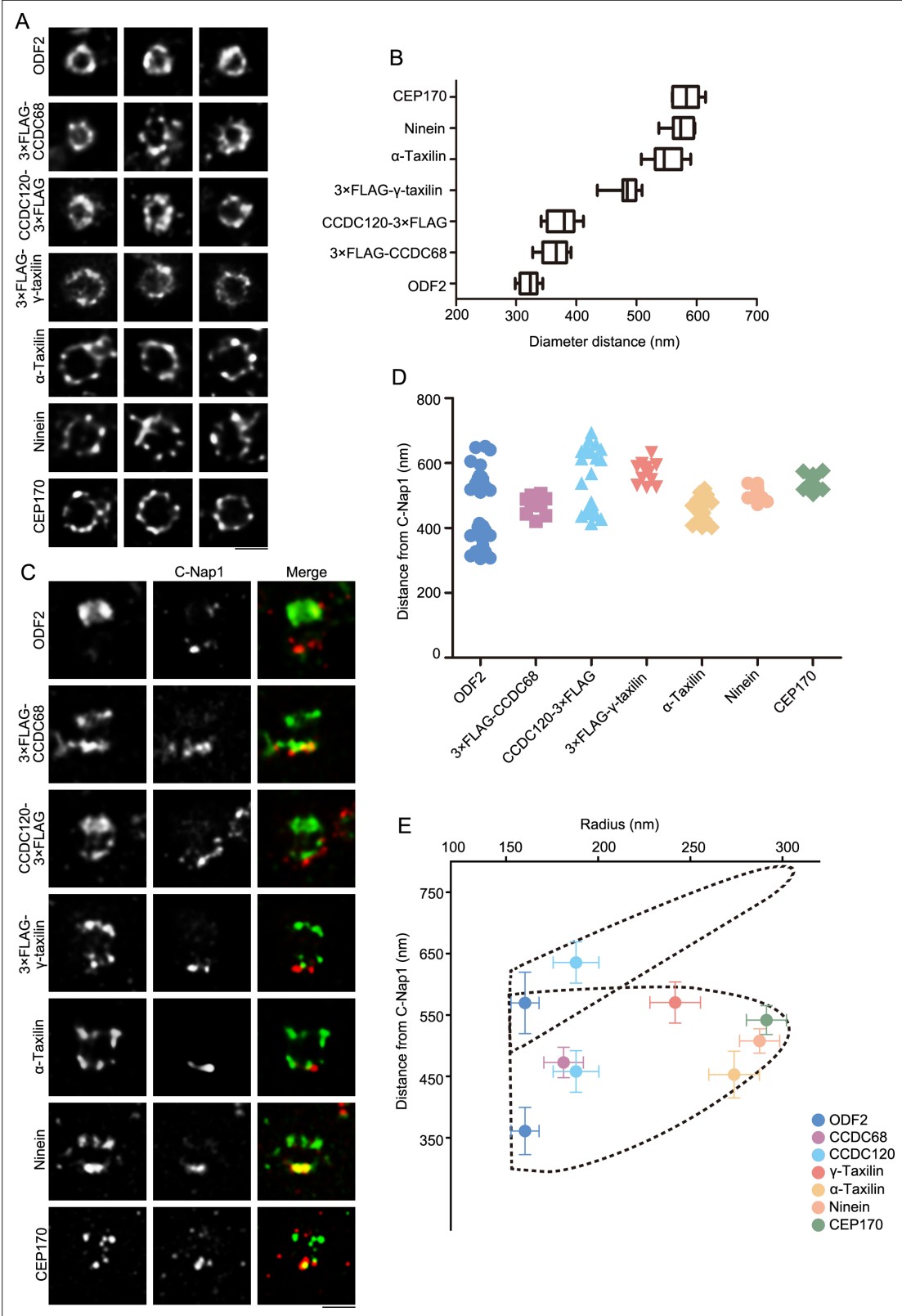

**Figure 2.** Specific localizations of subdistal appendage (SDA) proteins, including α-taxilin and γ-taxilin. (**A**) Representative simulated emission depletion (STED) super-resolution images showing top views of the subdistal appendage (SDA) protein distribution patterns. For ODF2, α-taxilin, ninein and CEP170, RPE-1 cells were immunostained with appropriate antibodies. For CCDC68, CCDC120 and γ-taxilin, RPE-1 cells overexpressed with 3×FLAG-tagged CCDC68, CCDC120 or γ-taxilin full-length were immunostained with FLAG antibody. Scale bar, 500 nm. (**B**) Diameter analysis of SDA proteins

*Figure 2 continued on next page*

*Figure 2 continued*

showing ring size diameter. Data are Mean ± SD. n ≥ 7, box = 25th and 75th percentiles. (**C**) Representative two-color STED super-resolution images showing side view of the SDA proteins (green) and the centriole proximal end protein C-Nap1 (red). Scale bar, 500 nm. (**D**) A scatter plot describing the distance of SDA proteins relative to C-Nap1. Data are Mean ± SD. n ≥ 11, box = 25th and 75th percentiles. (**E**) Relative localization of SDA proteins in radial and lateral directions of the mother centriole. The upper dotted lines reveal the slanted arrangement of distal appendage (DA) and the lower dotted lines represent the triangular SDA structure, respectively. Data are Mean ± SD.

The online version of this article includes the following source data for figure 2:

**Source data 1.** The diameter of subdistal appendage (SDA) proteins, including α-taxilin and γ-taxilin.

**Source data 2.** The longitudinal positions of subdistal appendage (SDA) proteins, including α-taxilin and γ-taxilin.

α-taxilin or γ-taxilin in HEK-293T cells (*Figure 3—figure supplement 1G-J*). Previously, TCHP was reported to reside at the SDA midzone and to interact with ODF2 and ninein (*Ibi et al., 2011*). Here, endogenous TCHP did not interact with either α-taxilin or γ-taxilin in HEK-293T cells (*Figure 3—figure supplement 1K*). Therefore, the proper localization of α-taxilin and γ-taxilin at the SDA depends on ODF2, but not on CCDC68, CCDC120, or TCHP.

## α-Taxilin and γ-taxilin form a complex at the SDA via their coiled-coil domains

We then moved on to determine the relationship between α-taxilin and γ-taxilin, which belong to the taxilin family and each possesses a coiled-coil domain in their middle regions (*Nogami et al., 2004*). First, we detected an interaction between those two proteins after immunoprecipitation in HEK-293T cells (*Figure 4A*). So, we then generated *α-taxilin* or *γ-taxilin* knockout (KO) RPE-1 cells using the CRISPR-Cas9 approach (*Figure 4—figure supplement 1A-F* and *Figure 4B–C*). Compared with that of wild-type (WT) cells, α-taxilin fluorescence intensity was less at the *γ-taxilin* KO cell centrosomes (*Figure 4D*). Conversely, *α-taxilin* depletion did not affect centrosomal γ-taxilin fluorescence intensity (*Figure 4D*). Mapping with ectopically expressed γ-taxilin and α-taxilin truncated mutants indicated that they interact with each other via their coiled-coil domains (*Figure 4E–H*). This was further confirmed via an in vitro binding assay using a purified bacteria-produced MBP-fused γ-taxilin M region (153–464 aa) and a GST-tagged α-taxilin M region (186–491 aa) (*Figure 4I*). These data suggest that γ-taxilin directly recruits α-taxilin to centrosomes via its coiled-coil domain.

## α-Taxilin recruits CEP170 to SDAs

The SDA marker proteins ninein and CEP170 occupy the peripheral region of the SDA structure (*Huang et al., 2017*; *Chong et al., 2020*), and based on the SDA protein localization pattern, α-taxilin is located upstream of those two proteins (*Figure 2A*). So, we investigated whether α-taxilin is involved in recruiting ninein and CEP170 to the SDAs. An endogenous immunoprecipitation assay showed that α-taxilin interacted with CEP170 but not with ninein (*Figure 4A*). Also, α-taxilin and CEP170 co-localized at both the SDAs and the proximal ends, as observed with STED nanoscopy (*Figure 5A*). To identify which segment of α-taxilin was responsible for that association, we overexpressed truncated mutants of HA-tagged α-taxilin in HEK-293T cells and mapped the mutants' interactions with CEP170. CEP170 interacted with α-taxilin in cells overexpressing full-length α-taxilin, as well as the M (186–491 aa) region and C-terminus (491–546 aa) deletion mutant constructs (ΔC), but not with the N-terminus construct (1–185 aa) (*Figure 5B–C*). This suggests that the α-taxilin M region, but not the N- or C-terminal regions, is required for its interaction with CEP170. An in vitro binding assay using GST-tagged α-taxilin-M and 3×FLAG-CEP170 purified from bacteria and HEK-293T cells (*Figure 5D*), respectively, suggesting that α-taxilin may directly bind CEP170.

Next, we examined the effect of α-taxilin depletion on CEP170's centrosomal localization and detected a significant decrease in CEP170 fluorescence intensity in RPE-1 cells treated with α-taxilin siRNA, compared with control siRNA (*Figure 5E–F*). Additionally, overexpressed siRNA-resistant full-length α-taxilin could rescue the CEP170 fluorescence intensity at the centrosomes, whereas the α-taxilin deletion mutant (ΔM2) lacking the region responsible for its SDA localization (*Figure 1G and I*) failed (*Figure 5E-F*). These results indicate that α-taxilin participates with CEP170 in SDA assembly.

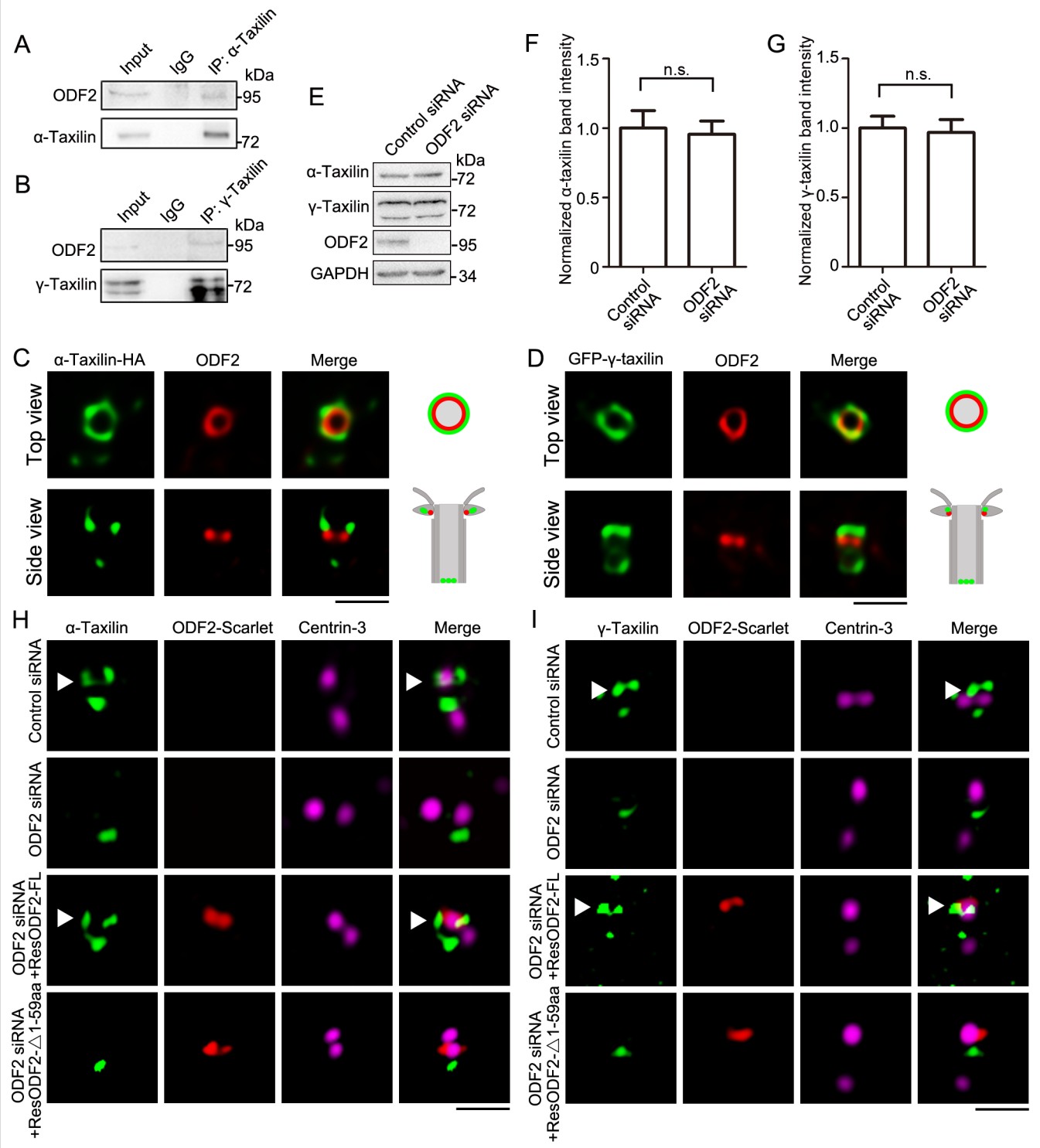

**Figure 3.** ODF2 is responsible for α-taxilin and γ-taxilin localization at the subdistal appendage (SDA). (**A**) Immunoblots of the endogenous immunoprecipitation (IP) assay of ODF2 and α-taxilin using anti-α-taxilin antibody in lysates of RPE-1 cells. IgG was the control. (**B**) Immunoblots of the endogenous IP assay of ODF2 and γ-taxilin using anti-γ-taxilin antibody in lysates of RPE-1 cells. (**C**) Simulated emission depletion (STED) images of RPE-1 cells immunostained with α-taxilin-HA (green) and ODF2 (red). Scale bar, 500 nm. (**D**) STED images of immunostained ODF2 (red) in RPE-1 cells transfected with GFP-γ-taxilin (green). The cartoons to the right of each set of images in (**C**) and (**D**) graphically depict the merge images. Scale bar, 500 nm. (**E**) Immunoblots of α-taxilin and γ-taxilin protein levels in control- or ODF2-siRNA treated RPE-1 cells. GAPDH was the loading control. (**F–G**) Comparisons of α-taxilin (**F**) and γ-taxilin (**G**) band intensities in (**E**). Data are Mean ± SEM. Statistical significance was determined by two-tailed Student's *t*-tests of six independent experiments. n.s., not significant. (**H**) Structured illumination microscopy (SIM) images of immunostained α-taxilin

*Figure 3 continued on next page*

*Figure 3 continued*

(green) and centrin-3 (magenta) in control- or ODF2-siRNA treated RPE-1 cells and those cells rescued by transfection with either siRNA-resistant scarlet-tagged ODF2 full-length (FL) or the 1–59 aa ODF2 deletion mutant (red). Scale bar, 1 µm. (**I**) SIM images of immunostained γ-taxilin (green) and centrin-3 (magenta) in control- or ODF2-siRNA treated RPE-1 cells and those cells rescued by transfection with either siRNA-resistant scarlet-tagged ODF2 FL or the 1–59 aa ODF2 deletion mutant (red). Scale bar, 1 µm. Arrowheads in (**H**) and (**I**) show α-taxilin and γ-taxilin SDA localizations, respectively.

The online version of this article includes the following source data and figure supplement(s) for figure 3:

**Source data 1.** Data of normalized α-taxilin band intensity in control- and ODF2-siRNA treated RPE-1 cells (Data provided as Mean ± SEM).

**Source data 2.** Data of normalized γ-taxilin band intensity in control- and ODF2-siRNA treated RPE-1 cells (Data provided as Mean ± SEM).

**Source data 3.** Full immunoblots labeled and unlabeled for *Figure 3A, B and E*.

**Figure supplement 1.** The α-taxilin and γ-taxilin assembly at the centrosome.

**Figure supplement 1—source data 1.** Data of normalized α-taxilin fluorescence intensity at the centrosome of control- and ODF2-siRNA treated RPE-1 cells (Data provided as Mean ± SEM).

**Figure supplement 1—source data 2.** Data of normalized γ-taxilin fluorescence intensity at the centrosome of control- and ODF2-siRNA treated RPE-1 cells (Data provided as Mean ± SEM).

**Figure supplement 1—source data 3.** Data of normalized α-taxilin fluorescence intensity at the centrosome of control- and CCDC68-siRNA treated RPE-1 cells (Data provided as Mean ± SEM).

**Figure supplement 1—source data 4.** Data of normalized γ-taxilin fluorescence intensity at the centrosome of control- and CCDC68-siRNA treated RPE-1 cells (Data provided as Mean ± SEM).

**Figure supplement 1—source data 5.** Data of normalized α-taxilin fluorescence intensity at the centrosome of control- and CCDC120-siRNA treated RPE-1 cells (Data provided as Mean ± SEM).

**Figure supplement 1—source data 6.** Data of normalized γ-taxilin fluorescence intensity at the centrosome of control- and CCDC120-siRNA treated RPE-1 cells (Data provided as Mean ± SEM).

**Figure supplement 1—source data 7.** Full immunoblots labeled and unlabeled for *Figure 3—figure supplement 1G-K*.

## α-Taxilin and γ-taxilin are responsible for SDA structural integrity and centrosome microtubule anchoring during interphase

To investigate the functions of α-taxilin and γ-taxilin at the SDA, we analyzed transmission electron microscope (TEM) images of centrioles in WT, *α-taxilin* KO and *γ-taxilin* KO RPE-1 cells (*Figure 6A*; *Figure 6—figure supplement 1A*). In WT RPE-1 cells, the side view showed that SDA stems were beside centriole wall and that they occupied a wide range at the subdistal position of the mother centriole (*Figure 6A*). The TEM images showed that a relatively narrower range of SDAs with smaller sizes appeared in *α-taxilin* KO RPE-1 cells and the smallest in *γ-taxilin* KO RPE-1 cells (*Figure 6A*; *Figure 6—figure supplement 1A*). These results suggest an indispensable role of α-taxilin and γ-taxilin in maintaining SDA integrity.

Since SDAs serve mainly as microtubule anchoring sites at the centrosome, we then examined whether α- and γ-taxilin are involved in microtubule organization. We began with a microtubule regrowth assay using *α-taxilin* KO or *γ-taxilin* KO RPE-1 cells (*Figure 4—figure supplement 1A-F* and *Figure 4B–C*). After ice-induced microtubule depolymerization, an immunofluorescence assay detected microtubule dynamics after rewarming. RPE-1 cells were fixed at different periods of time (0, 5, and 10 min), during which microtubules formed arrays radiating from the centrosomes (*Figure 6B–E*). The centrosomal microtubule aster in normal RPE-1 cells was visible 5 min after rewarming; and at 10 min, an extensive array of microtubules centering on the centrosome had formed (*Figure 6*, *B and D*). However, the microtubule asters were significantly impaired in the *α-taxilin* and *γ-taxilin* KO RPE-1 cells and the microtubule array densities were obviously less (*Figure 6*, *B and D*). To confirm this phenomenon, we further conducted microtubule regrowth assays in control and in α-taxilin and γ-taxilin siRNA-depleted RPE-1 cells. Those results displayed compromised microtubule reformation in the experimental cells, compared with that in the control cells (*Figure 6—figure supplement 1B-E*). The compromised microtubule regrowth, triggered by depletion of α-taxilin or γ-taxilin, could be partly rescued by overexpression of 3×FLAG-tagged full-length α-taxilin or γ-taxilin, but not by the M2 region (*Figure 1G–H*) deletion mutants of either taxilin (*Figure 6B–E*). This suggests that the SDA localization of both α-taxilin and γ-taxilin have indispensible roles in controlling microtubule anchoring at the interphase centrosome.

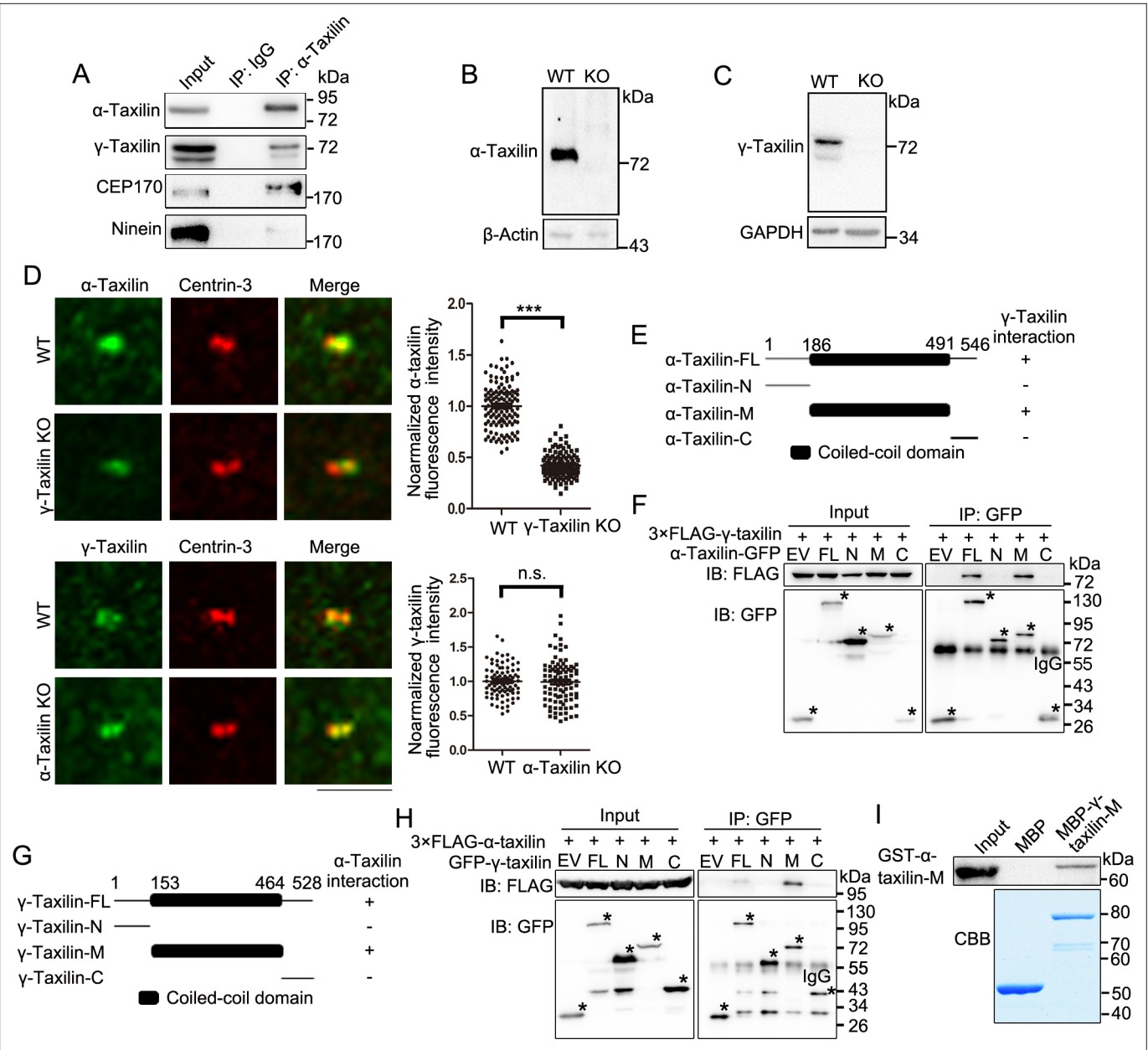

**Figure 4.** γ-Taxilin interacts with α-taxilin via their coiled-coil domains. (**A**) Endogenous immunoprecipitation (IP) assays of α-taxilin with γ-taxilin, CEP170, and ninein using anti-α-taxilin antibody in lysates of HEK-293T cells and were immunoblotted with the indicated antibodies. IgG was the control. (**B–C**) Immunoblots showing knockout of α-taxilin (**B**) and γ-taxilin (**C**) in RPE-1 knockout (KO) cells. β-Actin or GAPDH was used as loading controls. (**D**) Confocal images of immunostained α-taxilin (green) in wild-type (WT) and *γ-taxilin* KO RPE-1 cells, as well as immunostained γ-taxilin (green) in WT and *γ-taxilin* KO RPE-1 cells. Immunostained Centrin-3 (red) was used as a centrosome marker. Data were analyzed by two-tailed Student's *t*-tests with three experimental replicates and expressed as Mean ± SEM. n > 80; ***, p < 0.001; n.s., not significant. (**E**) Schematic showing the interactions of full-length (FL) α-taxilin and its truncated mutants (N terminus, [M] Middle, and C terminus) with γ-taxilin. +, positive; −, negative. (**F**) Lysates of HEK-293T cells co-overexpressing GFP-tagged FL α-taxilin or its truncated mutants from (**E**) with 3× FLAG-γ-taxilin were subjected to IP and immunoblotted (IB) with anti-GFP and anti-FLAG antibodies. EV, empty vector. The stars marked the indicated bands in (**E**). (**G**) Schematic showing FL γ-taxilin and its truncated mutants' interactions with α-taxilin. (**H**) Lysates of HEK-293T cells co-overexpressing GFP-tagged FL γ-taxilin or its truncated mutants from (**G**) with 3×FLAG-α-taxilin were subjected to IP with anti-GFP and IB with anti-GFP and anti-FLAG antibodies. The stars marked the indicated bands in (**G**). (**I**) In vitro binding assay of MBP-γ-taxilin-M from (**G**) (expressed in *E. coli*, purified, and stained with Coomassie brilliant blue [CBB]) with GST-α-taxilin-M from (**E**) (expressed in *E. coli* and pulled down and detected by IB using the GST antibody).

The online version of this article includes the following source data and figure supplement(s) for figure 4:

**Source data 1.** Centrosomal α-taxilin fluorescence intensities in wildtype (WT) and *γ-taxilin* knockout (KO) RPE-1 cells (Data provided as Mean ± SEM).

*Figure 4 continued on next page*

Figure 4 continued

**Source data 2.** Centrosomal γ-taxilin fluorescence intensities in wildtype (WT) and *α-taxilin* knockout (KO) RPE-1 cells (Data provided as Mean ± SEM).

**Source data 3.** Full immunoblots labeled and unlabeled for *Figure 4A, B, C, F and H*.

**Figure supplement 1.** Characterizations of α-*taxilin* or γ-*taxilin* wildtype (WT) and knockout (KO) RPE-1 cells.

**Figure supplement 1—source data 1.** Percentage of cells with centrosome separated over 2 μm in WT and *α-taxilin* KO RPE-1 cells (Data provided as Mean ± SEM).

**Figure supplement 1—source data 2.** Percentage of cells with centrosome separated over 2 μm in WT and *γ-taxilin* KO RPE-1 cells (Data provided as Mean ± SEM).

The centrosome acts as an MTOC by controlling microtubule nucleation and anchoring. γ-Tubulin, as a member of the γ-TuRC complex, plays a major role in microtubule nucleation (*Schatten and Sun, 2018*). To determine the causes of the compromised microtubule reformation observed in both α- and γ-taxilin depleted cells, we examined γ-tubulin at the interphase centrosome in control-, α-taxilin, and γ-taxilin-siRNA treated RPE-1 cells. The γ-tubulin fluorescence intensity at the interphase centrosome remained unchanged following both α-taxilin and γ-taxilin siRNA-induced depletion (*Figure 6—figure supplement 1F-G*), thus suggesting that microtubule nucleation may not be affected.

### α-Taxilin and γ-taxilin are required for proper spindle orientation during metaphase

In proliferating cells, centrosomes act as spindle poles when cells enter mitosis. Therefore, we investigated whether α-taxilin or γ-taxilin loss affected spindle formation. For this, we generated *α-taxilin* KO and *γ-taxilin* KO HeLa cells using the CRISPR-Cas9 approach (*Figure 7—figure supplement 1A-F* and *Figure 7A–B*). Although bipolar spindle formation was not affected in both *α-* and *γ-taxilin* KO cells (*Figure 7C*), changes in the orientation of the spindle to the substratum were obvious (*Figure 7C–E*). The average spindle angle increased from 5.13 (SE 0.43)° in WT cells to 11.92 (0.95)° in *α-taxilin* KO cells, and from 4.94 (0.46)° in WT cells to 10.94 (0.93)° in *γ-taxilin* KO cells (*Figure 7D–E*). Spindle misorientations were rescued by overexpression of either 3× FLAG-tagged full-length α-taxilin or γ-taxilin, as well as by the similarly tagged α-taxilin M3 deletion mutant or the γ-taxilin M1 deletion mutant (*Figure 1G–H*), but not by the M2 deletion mutant for each taxilin (*Figure 7C–E*; *Figure 1G–H*). These results confirm α- and γ-taxilin roles in maintaining proper spindle orientation during metaphase.

Finally, because astral microtubule loss causes spindle misorientation (*Hung et al., 2016a*), we examined possible losses by staining the plus-end microtubule binding protein EB1 (*Honnappa et al., 2009*) in WT and in *α-taxilin* KO and *γ-taxilin* KO HeLa cells. While the WT's aster microtubule length was 3.09 (SE 0.07) μm, those lengths in the *α-taxilin* (2.20 [0.07] μm) and *γ-taxilin* (2.19 [0.07] μm) KO HeLa cells were significantly less (*Figure 7F–G*). These data suggest that the changes in spindle orientation observed in *α-taxilin* and *γ-taxilin* KO HeLa cells are likely caused by decreased astral microtubule length.

## Discussion

SDAs are conserved structures located at the subdistal end of the mother centriole, and their formation, together with that of the DAs, marks centriole maturity (*Uzbekov and Alieva, 2018*). They play important roles in microtubule organization and spindle arrangement and participate in various biological processes such as cell division and cell differentiation (*Hall and Hehnly, 2021*). Here, we applied APEX2-mediated proximity-labeling to centrosome proteomics by using CCDC68 and CCDC120 as baits (*Hung et al., 2016b*; *Huang et al., 2017*). The results show a variety of possible proteins, some of which are known centrosomal proteins like γ-tubulin and γ-taxilin, and several new centrosome-localized proteins are also identified (*Figure 1—figure supplement 1A-B*), such as α-taxilin, DACT1, NCAPH2, DRG2, and SMAP2. Among these proteins, α-taxilin and γ-taxilin are both located at the SDA and the proximal end of mother centriole (*Figures 1–3*). However, other already known SDA proteins, such as ODF2, ninein and CEP170, are not included in the datasets, probably because of their relatively low levels in the cells (for ODF2, *Figure 3A–B*) or their relatively longer distances from CCDC68 and CCDC120 (for ninein and CEP170, *Figure 2*).

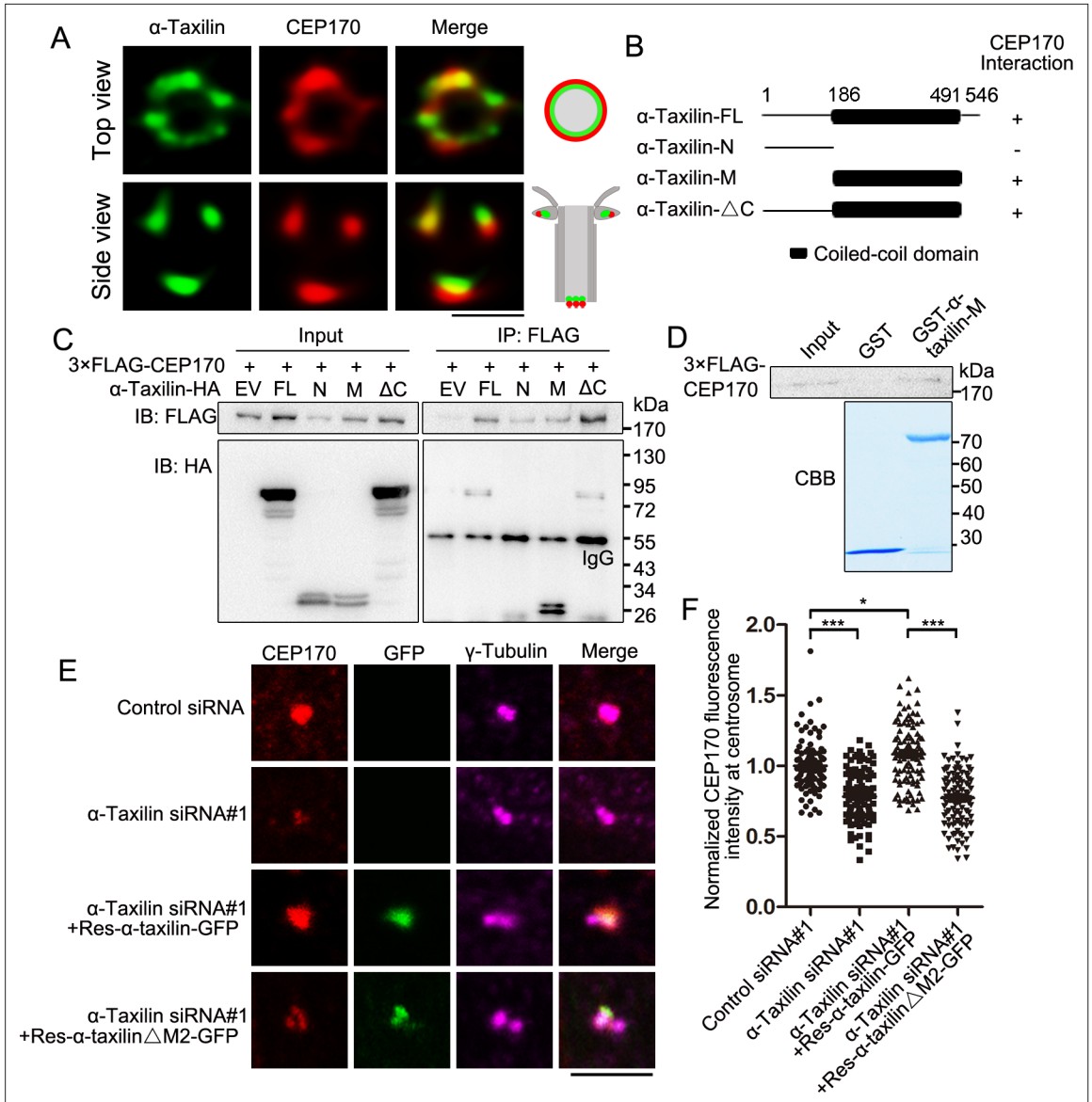

**Figure 5.** α-Taxilin recruits CEP170 to the subdistal appendage (SDA). (**A**) Simulated emission depletion (STED) images of RPE-1 cells immunostained with α-taxilin (green) and CEP170 (red). Scale bar, 500 nm. The cartoons to the right of the images graphically depict the merge images. (**B**) Schematic showing the full-length (FL) α-taxilin and the truncated mutants (N terminus, [M] Middle, deleted C terminus). +, positive; −, negative. (**C**) Lysates of HEK-293T cells co-overexpressing HA-tagged α-taxilin-FL or the indicated truncated mutants in (**B**) with 3×FLAG-CEP170 were immunoprecipitated (IP) with anti-FLAG and immunoblotted (IB) with anti-HA and anti-FLAG antibodies. EV, empty vector. (**D**) In vitro binding of GST-α-taxilin-M from (**B**) (expressed in *E. coli* and purified) with 3×FLAG-CEP170 (expressed in HEK-293T cells and purified). The GST-α-taxilin-M was stained with Coomassie brilliant blue (CBB) and the 3×FLAG-CEP170 was pulled down and IB using FLAG antibody. (**E**) Confocal images of immunostained CEP170 (red) and γ-tubulin (magenta) in control- or α-taxilin-siRNA treated RPE-1 cells, and those cells rescued with siRNA-resistant GFP-tagged α-taxilin-FL or the α-taxilin M2 deletion mutant (Δ261–300 aa) (green). Scale bar, 5 µm. (**F**) Comparisons of CEP170 fluorescence intensities at the centrosomes in (**E**). Statistical significance was determined with one-way ANOVA with three replicates. Data are Mean ± SEM. n > 100; *, p < 0.05; ***, p < 0.001; n.s., not significant.

The online version of this article includes the following source data for figure 5:

**Source data 1.** Data of centrosomal CEP170 fluorescence intensity in control and α-taxilin siRNA treated RPE-1 cells, and rescued by overexpressed full-length α-taxilin or the α-taxilin M2 deletion mutant (Data provided as Mean ± SEM).

**Source data 2.** Full immunoblots labeled and unlabeled for *Figure 5C and D*.

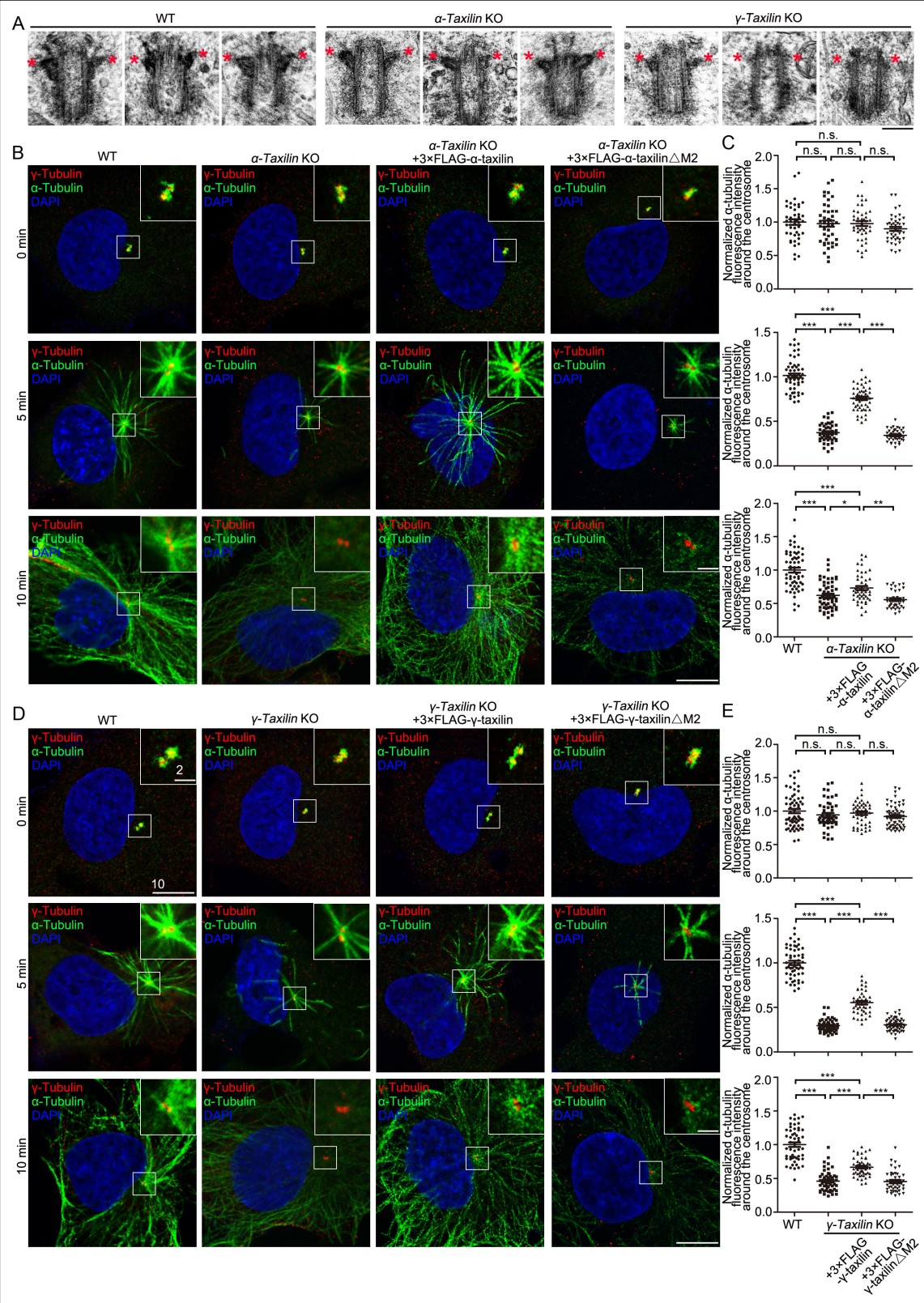

**Figure 6.** α-*Taxilin* and γ-*taxilin* knockout (KO) inhibits microtubule reformation in RPE-1 cells after cold depolymerization. (**A**) Transmission electron microscope (TEM) images of the mother centriole of wild-type (WT), α-*taxilin* and γ-*taxilin* KO RPE-1 cells. Red stars marked subdistal appendages. Scale bar, 200 nm. (**B**) Confocal images of microtubule reformation at 0, 5, and 10 min after rewarming. Wild-type (WT), α-*taxilin* KO RPE-1 cells, and those cells rescued with 3×FLAG-tagged full-length (FL) α-taxilin and the α-taxilin M2 deletion mutant (△261–300 aa, shown in **Figure 1G**) were

*Figure 6 continued on next page*

*Figure 6 continued*

immunostained with antibodies against α-tubulin (green) and centrosome marker γ-tubulin (red). DNA was stained with DAPI (blue). Scale bars, 5 µm. (**C**) Comparisons of α-tubulin fluorescence intensities around the centrosome in (**B**). Statistical significance was determined by one-way ANOVA. n > 30. Data are mean ± SEM. *, p < 0.05; **, p < 0.01; ***, p < 0.001; n.s., not significant. (**D**) Confocal images of microtubule reformation at 0, 5, and 10 min after rewarming. WT, *γ-taxilin* KO RPE-1 cells, and those cells rescued with 3×FLAG-tagged γ-taxilin-FL and the γ-taxilin M2 deletion mutant (△271–317 aa, shown in *Figure 1H*) were immunostained with antibodies against α-tubulin (green) and centrosome marker γ-tubulin (red). DNA was stained with DAPI (blue). Scale bars, 5 µm. (**E**) Statistical analyses of α-tubulin fluorescence intensity around the centrosome in (**D**). Statistical significance of α-tubulin fluorescence intensity at 0, 5, and 10 min was determined by one-way ANOVA. n > 40. Data are Mean ± SEM. *, p < 0.05; **, p < 0.01; ***, p < 0.001; n.s., not significant.

The online version of this article includes the following source data and figure supplement(s) for figure 6:

**Source data 1.** Data of normalized centrosomal α-tubulin fluorescence intensity in wild-type (WT), *α-Taxilin* knockout (KO) RPE-1 cells, and cells rescued by overexpressed 3×FLAG-α-taxilin or 3× FLAG-α-taxilin△M2 (Data provided as Mean ± SEM).

**Source data 2.** Data of normalized centrosomal α-tubulin fluorescence intensity in wild-type (WT), *γ-Taxilin* knockout (KO) RPE-1 cells, and rescued by overexpression of 3×FLAG-γ-Taxilin or 3×FLAG-γ-Taxilin△M2 (Data provided as Mean ± SEM).

**Figure supplement 1.** α-Taxilin and γ-taxilin depletion inhibits microtubule reformation in RPE-1 cells after cold depolymerization.

**Figure supplement 1—source data 1.** Data of normalized areas of SDAs in wild-type (WT), *α-taxilin* and *γ-taxilin* KO RPE-1 cells (Data provided as Mean ± SEM).

**Figure supplement 1—source data 2.** Data of normalized centrosomal α-tubulin fluorescence intensity in control-, α-taxilin-, or γ-taxilin-siRNA treated RPE-1 cells (Data provided as Mean ± SEM).

**Figure supplement 1—source data 3.** Data of normalized centrosomal γ-tubulin fluorescence intensity in control-, α-taxilin-, or γ-taxilin-siRNA treated RPE-1 cells (Data provided as Mean ± SEM).

The SDA is a comprehensive structure that consists of a spreading radial distribution of concentric proteins, and SDA assembly follows a hierarchical relationship based on protein distributions (*Chong et al., 2020*). Comparing SDA protein diameters, those of α- and γ-taxilin are larger than those of CCDC68 and CCDC120, but smaller than those of ninein and CEP170. In the longitudinal position, γ-taxilin is relatively higher while α-taxilin is relatively lower than the other SDA proteins. Associated with their relative localization with other SDA proteins, they are supposed to reside in the middle SDA zone (*Figure 8A–B*).

Among the currently known SDA components, ODF2 is the most closely located to the centriolar microtubules (*Figure 8C*) and it likely recruits TCHP to assist ninein assembly at the SDA (*Ibi et al., 2011*). The ODF2 1–60 aa N-terminus sequence is responsible for its association with CCDC120 (*Huang et al., 2017*). However, this association may not be direct, as a yeast two-hybrid assay indicated. Besides, CCDC68 lies between ODF2 and CEP170, and how CCDC68 is recruited to the SDA is yet unknown (*Huang et al., 2017*). Our data suggest that ODF2 recruits α- and γ-taxilin, similarly to how it recruits CCDC120 (*Huang et al., 2017*). Since the ODF2 ring is much smaller than those of both α- and γ-taxilin, most likely the interactions between ODF2 and the taxilins are indirect (*Figure 8A–B*). Although TCHP, CCDC68, and CCDC120 lie between ODF2 and the taxilins (*Figure 8A–B*), no interactions among taxilins with those SDA proteins, except for ODF2, are detected by immunoprecipitation assays. Therefore, ODF2 recruits α-taxilin and γ-taxilin through a new pathway (*Figure 8C*).

The functions of SDA components are critical for understanding this structure. Ninein is responsible not only for microtubule nucleation by docking the γ-TuRC at the centrosomes (*Delgehyr et al., 2005*), but also for forming a microtubule-anchoring complex with CEP170 at the SDA periphery (*Pizon et al., 2020*). CCDC68 and CCDC120 also participate in microtubule anchoring (*Huang et al., 2017*). Similarly, compromised microtubule reformation following ice-depolymerization was observed in both α-taxilin and γ-taxilin depleted cells, likely because of depressed microtubule anchoring ability, as γ-tubulin intensity at the centrosome is not influenced by either α-taxilin or γ-taxilin siRNA knockdown. Therefore, our data suggest that α-taxilin and γ-taxilin serve as microtubule anchoring regulators at the centrosomes, functioning via a new pathway that is independent of CCDC68, CCDC120, and ninein.

In addition to acting as microtubule anchoring centers during interphase, SDAs also function as spindle regulators during mitosis. CEP170 interacts with centrosome-associated kinesins such as KIF2A, KIF2C, KIFC3, and spindle microtubule-associated KIF2B (*Welburn and Cheeseman, 2012*; *Maliga et al., 2013*), which regulates spindle assembly and cell morphology. In our study, both α-taxilin and γ-taxilin depletion result in decreased astral microtubule length and increased spindle misorientation.

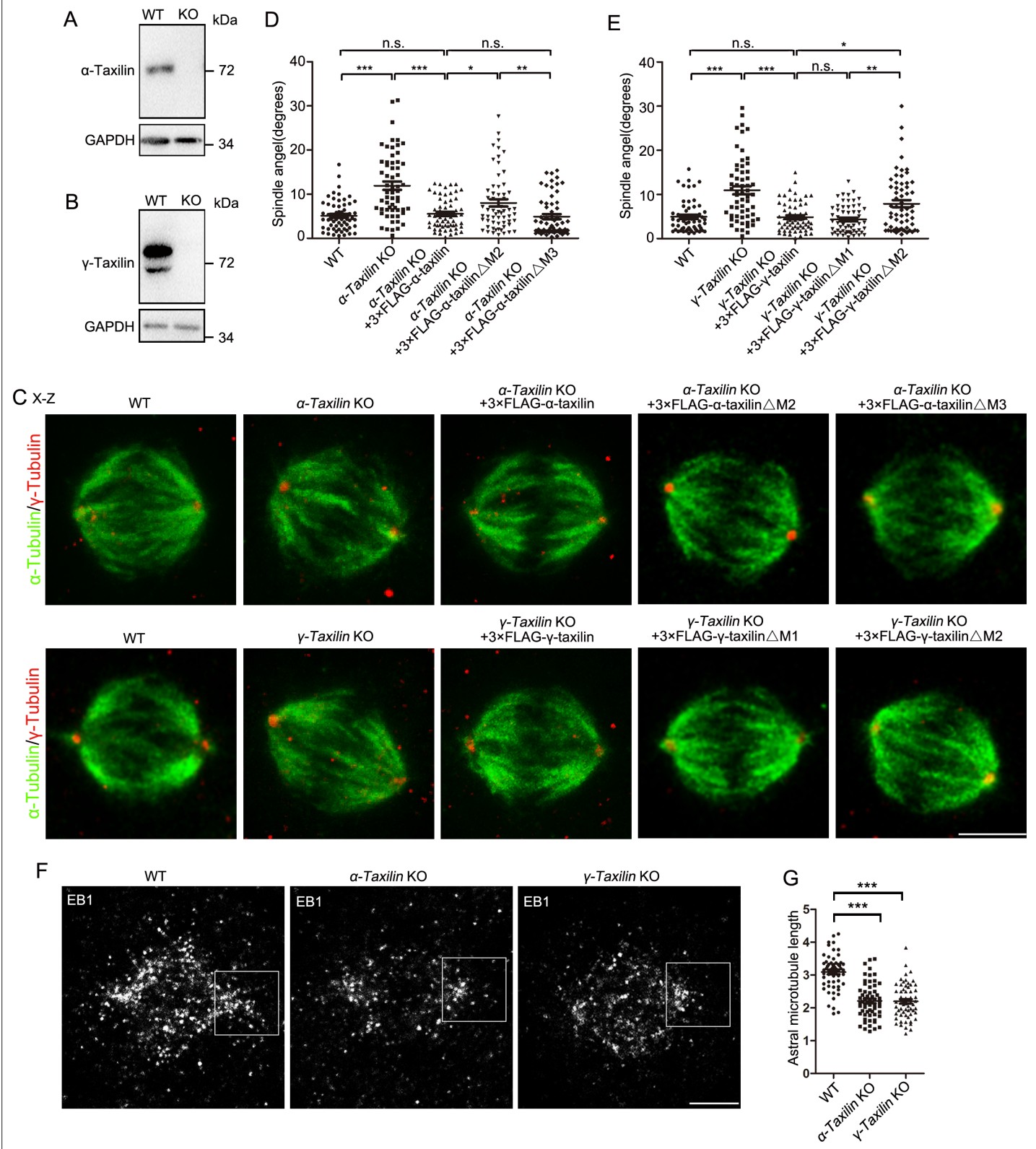

**Figure 7.** α-*Taxilin* and γ-*taxilin* knockout (KO) increases spindle angle orientation during mitosis. (**A–B**) Immunoblots showing wild-type (WT) and KO α-taxilin (**A**) and γ-taxilin (**B**) in HeLa cells. GAPDH was the loading control. (**C**) Representative orthogonal views (**x–z**) of metaphase HeLa cells stained for microtubules (α-tubulin, green) and spindle poles (γ-tubulin, red). Views showed WT, α-*taxilin*, or γ-*taxilin* KO cells, and cells overexpressed with 3×FLAG-tagged full-length α-taxilin or γ-taxilin and the indicated deletion mutants (Δ) described in *Figure 1G and H*, respectively. Scale bars,

*Figure 7 continued on next page*

*Figure 7 continued*

10 μm. (**D**) Comparisons of raw spindle angles in WT, α-taxilin KO HeLa cells, and cells overexpressed with 3×FLAG-tagged full-length α-taxilin or the indicated deletion mutants (Δ). Data are represented as Mean ± SEM. Statistical significance was determined by one-way ANOVA of three individual experiments. n ≥ 60. *, $p < 0.05$; **, $p < 0.01$; ***, $p < 0.001$; n.s., not significant. (**E**) Comparisons of raw spindle angles in WT, γ-taxilin KO HeLa cells, and cells overexpressed with 3×FLAG-tagged full-length γ-taxilin or the indicated deletion mutants (Δ). Data are represented as Mean ± SEM. Statistical significance was determined by one-way ANOVA of three individual experiments. n ≥ 60. *, $p < 0.05$; **, $p < 0.01$; ***, $p < 0.001$; n.s., not significant. (**F**) Confocal images of aster microtubules immunostained with EB1 in WT, α-taxilin, or γ-taxilin KO HeLa cells. Scale bars, 5 μm. (**G**) Comparisons of the aster microtubule lengths in (**F**). Data are mean (SEM). Statistical significance was determined by one-way ANOVA of three replicates. n ≥ 60. ***, $p < 0.001$.

The online version of this article includes the following source data and figure supplement(s) for figure 7:

**Source data 1.** Spindle angles of wild-type (WT), *α-taxilin* knockout (KO) HeLa cells, and cells overexpressed with indicated *α-taxilin* full-length and deletion mutants (Data provided as Mean ± SEM).

**Source data 2.** Spindle angles of wild-type (WT), *γ-taxilin* knockout (KO) HeLa cells, and cells overexpressed with indicated *γ-taxilin* full-length and deletion mutants (Data provided as Mean ± SEM).

**Source data 3.** Astral microtubule length in wild-type (WT), *α-taxilin,* and *γ-taxilin* knockout (KO) HeLa cells (Data provided as Mean ± SEM).

**Source data 4.** Full immunoblots labeled and unlabeled for *Figure 7A, B*.

**Figure supplement 1.** Characterizations of α-taxilin and γ-taxilin WT and KO HeLa cells.

**Figure supplement 1—source data 1.** Percentage of cells with centrosome separated over 2 μm in WT and α-taxilin KO HeLa cells (Data provided as Mean ± SEM).

**Figure supplement 1—source data 2.** Percentage of cells with centrosome separated over 2 μm in WT and γ-taxilin KO HeLa cells (Data provided as Mean ± SEM).

As the hierarchical assembly of γ-taxilin, α-taxilin, and CEP170 is established, we speculate that α-taxilin and γ-taxilin regulate spindle orientation mainly by controlling CEP170's centrosomal localization. In addition to CEP170, the most upstream SDA protein, ODF2, is shown to regulate spindle orientation via microtubule organization and stability (*Hung et al., 2016a*). Spindle disorganization has been extensively correlated with cell differentiation, cancer and neurological diseases such as microcephaly and lissencephaly (*Noatynska et al., 2012*), as well as with tubular organ diseases (*Zhong and Zhou, 2017*). Whether α- and γ-taxilin are involved in development and diseases needs further investigation.

Our study mainly investigate the manner of assembly and functions of α-taxilin and γ-taxilin at the SDA, but the specific localization of those proteins, as revealed by super resolution microscopy, also shows proximal signals at centrioles (*Figure 1A–F*) that resembled ninein group patterns (*Mazo*

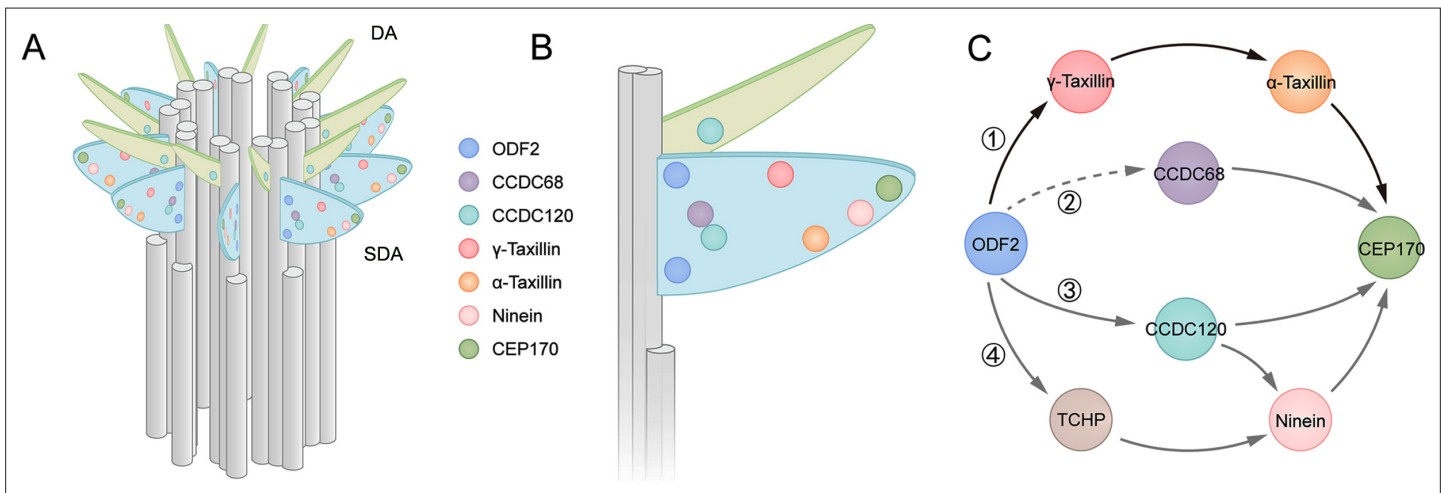

**Figure 8.** Models showing α-taxilin and γ-taxilin localization and assembly in the subdistal appendage (SDA). (**A**) A 3D model of a mother centriole, illustrating the localization of various SDA proteins, including ODF2, CCDC68, CCDC120, γ-taxilin, α-taxilin, ninein, and CEP170 in order of increasing diameter size. (**B**) Close view of the SDA structure from (**A**). (**C**) Model showing the hierarchical relationships of SDA proteins with ODF2 at the SDA root: (1) ODF2 recruits γ-taxilin to the SDA, which in turn hierarchically recruits α-taxilin and then CEP170; (2) CCDC68 is recruited to the SDA, which further recruits CEP170 (*Huang et al., 2017*); (3) ODF2 recruits ninein and CEP170 via CCDC120 (*Huang et al., 2017*); (4) ODF2 recruits ninein via TCHP (*Ibi et al., 2011*). DA, distal appendage.

*et al., 2016*). Previously, γ-taxilin is shown, by interacting with mitosis A-related kinase 2 A (Nek2A), to temporally mediate centrosome disjunction before the onset of mitosis (*Makiyama et al., 2018*). This is consistent with our finding that γ-taxilin knockout also resulted in centrosomal disjunction in RPE-1 and HeLa cells (*Figure 4—figure supplement 1C-D*; *Figure 7—figure supplement 1C-D*). Similarly, α-taxilin depletion in RPE-1 and HeLa cells also result in centrosomal disjunction (*Figure 4—figure supplement 1E-F*; *Figure 7—figure supplement 1E-F*) that resembles the γ-taxilin phenotype. Therefore, the proximal presence of α-taxilin and γ-taxilin at centrioles may work as centrosome linkers regulating centrosome cohesion, and their proximal end assembly rules need further study.

In conclusion, our results both show α-taxilin and γ-taxilin to be new SDA proteins and shed new light on how SDAs are assembled. However, other yet unknown components exist at the SDA upper zone and participate in SDA assembly, and they and their functions need to be identified. Additionally, how α-taxilin and γ-taxilin function in development and diseases warrants further investigation.

# Materials and methods

## Key resources table

| Reagent type (species) or resource | Designation | Source or reference | Identifiers | Additional information |
|---|---|---|---|---|
| Gene (*Homo-sapiens*) | CCDC68 | GenBank | NM_025214.3 | |
| Gene (*Homo-sapiens*) | CCDC120 | GenBank | NM_001163321.4 | |
| Gene (*Homo-sapiens*) | α-taxilin | GenBank | NM_175852.4 | |
| Gene (*Homo-sapiens*) | γ-taxilin | GenBank | XM_024452398.1 | |
| Gene (*Homo-sapiens*) | SWAP2 | GenBank | AY358944.1 | |
| Gene (*Homo-sapiens*) | DACT1 | GenBank | NM_016651.6 | |
| Gene (*Homo-sapiens*) | ODF2 | GenBank | NM_001242352.1 | |
| Cell line (*Homo-sapiens*) | HeLa | ATCC | CCL-2 | Cell line authenticated by ATCC |
| Cell line (*Homo-sapiens*) | U2OS | ATCC | HTB-96 | Cell line authenticated by ATCC |
| Cell line (*Homo-sapiens*) | HEK-293T | ATCC | CRL-3216 | Cell line authenticated by ATCC |
| Cell line (*Homo-sapiens*) | RPE-1 | ATCC | CRL-4000 | Cell line authenticated by ATCC |
| Antibody | α-taxilin (Rabbit polyclonal) | Proteintech | 27558–1-AP | IF(1:100), IB (1:1000) |
| Antibody | α-tubulin (Mouse monoclonal) | Sigma-Aldrich | T9026 | IF (1:1000) |
| Antibody | β-actin (Mouse monoclonal) | Cell Signaling Technology | 8H10D10 | IB (1: 10000) |
| Antibody | γ-taxilin (Rabbit polyclonal) | Proteintech | 22357–1-AP | IF(1:100), IB (1:1000) |
| Antibody | γ-tubulin (Rabbit polyclonal) | Sigma-Aldrich | T3559 | IF (1:200) |
| Antibody | Biotin (Mouse monoclonal) | Sigma-Aldrich | BN-34 | IB (1:500) |
| Antibody | Biotin-FITC (Goat polyclonal) | Sigma-Aldrich | F6762 | IF (1:100) |
| Antibody | Centrin-3 (Mouse monoclonal) | Abnova | ABN-H0000 1070-M01 | IF (1:200) |

*Continued on next page*

*Continued*

| Reagent type (species) or resource | Designation | Source or reference | Identifiers | Additional information |
|---|---|---|---|---|
| Antibody | CEP170 (Rabbit polyclonal) | Abcam | ab72505 | IF (1:100); IB (1:1000) |
| Antibody | CEP170 (Rabbit polyclonal) | Proteintech | 27325–1-AP | IF (1:100) |
| Antibody | CEP170 (Mouse monoclonal) | Invitrogen | 72-413-1 | IF (1:200) |
| Antibody | EB1 (Rabbit polyclonal) | Proteintech | 17717–1-AP | IF (1:100) |
| Antibody | FLAG (Mouse monoclonal) | Sigma-Aldrich | F3165 | IB (1:10000) |
| Antibody | GAPDH (Mouse monoclonal) | ABclonal | AC002 | IB (1:20000) |
| Antibody | GFP (Rabbit polyclonal) | Laboratory produced | | IB (1:10000) |
| Antibody | GST (Rabbit polyclonal) | Laboratory produced | | IB (1:500) |
| Antibody | HA (Mouse monoclonal) | Sigma-Aldrich | H9658 | IF (1:1000); IB (1:10000) |
| Antibody | ninein (Rabbit polyclonal) | Abcam | ab231181 | IF (1:100); IB (1:1000) |
| Antibody | ninein (Mouse monoclonal) | Invitrogen | 79.160–7 | IF (1:100) |
| Antibody | ODF2 (Rabbit polyclonal) | Proteintech | 12058–1-AP | IF (1:100); IB (1:1000) |
| Antibody | TCHP (Rabbit polyclonal) | ATLAS antibodies | HPA038638 | IB (1:1000) |
| Antibody | V5 (Mouse monoclonal) | Innovative Research | R960CUS | IF (1:1000); IB (1:10000) |
| Recombinant DNA reagent | α-taxilin gRNA (plasmid) | This paper | | gRNA (5'-GCAGTAGAA GCAGAAGGTCC-3') |
| Recombinant DNA reagent | γ-taxilin gRNA (plasmid) | This paper | | gRNA (5'-CTGTTCGA GCTTCTCCCCCA-3') |
| Software, algorithm | Image J | NIH (*Schneider et al., 2012*) | | |
| Software, algorithm | Prism 5 | GraphPad | | |

## Cell cultures and transfection

HeLa (ATCC, CCL-2), U2OS (ATCC, HTB-96), and HEK-293T (ATCC, CRL-3216) cells were purchased from ATCC (Manassas, VA, USA) and cultured in DMEM (GIBCO, Thermo Fisher Scientific, Waltham, MA, USA) with 10% FBS (CellMax, Lanzhou Minhai Bio-Engineering, Lanzhou, China), while RPE-1 cells (ATCC, CRL-4000) were cultured in DMEM/F12 (1:1) (GIBCO) with 10% FBS. The identity of the cell lines has been authenticated by ATCC with a 100% match. Cells were validated to be negative for mycoplasma contamination. All cells were incubated at 37 °C with a 5% $CO_2$ atmosphere. The HeLa, U2OS, and HEK-293T cells were transfected using polyethylenimine (#23966–1, Polysciences, Inc, Warrington, PA, USA), while the RPE-1 cells were transfected using Lipofectamine 3000 (#L3000-015, Invitrogen), both according to the manufacturer's instructions.

## Plasmid construction

To obtain CCDC68-V5-APEX2 and CCDC120-V5-APEX2 constructs, human CCDC68 (NM_025214.3) and CCDC120 (NM_001163321.4), respectively, were amplified by PCR from human cDNA and cloned into mito-V5-APEX2 (*Lam et al., 2015*). Mito-V5-APEX2 was a gift from Alice Ting (Addgene plasmid # 72480; http://n2t.net/addgene:72480; RRID: Addgene_72480). Human γ-taxilin (XM_024452398.1), SWAP2 (AY358944.1), and DACT1 (NM_016651.6) were amplified by PCR from human cDNA and cloned into pmNeonGreenHO-G (*Tanida-Miyake et al., 2018*), a gift from Isei Tanida (Addgene plasmid # 127912; http://n2t.net/addgene:127912; RRID: Addgene_127912). Human ODF2

(NM_001242352.1) and its deletion mutant (Δ1–59 aa) were amplified by PCR from human cDNA and cloned into pScarletN1, a gift from Oskar Laur (Addgene plasmid # 128060; http://n2t.net/addgene: 128060; RRID: Addgene_128060). Using PCR, α-taxilin (NM_175852.4) and its truncated segments were amplified from human cDNA and cloned into pcDNA3.1(+) (Invitrogen). Full-length α-taxilin and γ-taxilin were also cloned into p3×FLAG-CMV-14 (Sigma-Aldrich, St. Louis, MO, USA). Ninein-GFP, 3×FLAG-CEP170, and CCDC120-GFP were obtained from a previous study (*Huang et al., 2017*).

## APEX2-mediated proximal labeling of CCDC68 and CCDC120

Once CCDC68-V5-APEX2 and CCDC120-V5-APEX2 were constructed, a large-scale proteomic experiment was conducted in cultured HEK-293T cells in which both constructs were over-expressed for 24 hr. Following treatment with BP (final concentration of 50 µM) for 30 min and $H_2O_2$ (final concentration of 1 mM) for 1 min, the biotinylated proteins were enriched by Streptavidin-coated beads (Streptavidin Sepharose High Performance, #17-5113-01, Cytiva, Marlborough, MA, USA) for mass spectrographic analysis.

## Mass spectrometry

To identify proteins, the Coomassie-stained proteins bands of each sample were cut out of the gels and destained with a solution of 25 mM ammonium bicarbonate in 50% acetonitrile. After dithiothreitol reduction and iodoacetamide alkylation, the proteins were digested with porcine trypsin (Sequencing grade modified; Promega, Madison, WI, USA) overnight at 37 °C. The resulting tryptic peptides were extracted from the gel pieces by using 200 µl of acetonitrile (with 0.1% formic acid). The samples were dried in a vacuum centrifuge concentrator at 30 °C and re-suspended in 10 µl of 0.1% formic acid.

The peptides, resolved using 0.1% formic acid, were loaded into a trap column (Acclaim PepMap 100 75 µm × 2 cm nanoViper, $C_{18}$, 3 µm, 100 Å, Thermo Fisher Scientific) and connected to an analytical column (Acclaim PepMap RSLC 75 µm × 15 cm nanoViper, $C_{18}$, 2 µm, 100 Å, Thermo Fisher Scientific) on a nanoflow HPLC Easy-nLC 1200 system (Thermo Fisher Scientific) using a 75 min LC gradient at 280 nl/min. Buffer A consisted of 0.1% (v/v) formic acid in $H_2O$ and Buffer B consisted of 0.1% (v/v) formic acid in 80% acetonitrile. The gradient was set as follows: 4–8% B in 5 min, 8–20% B in 45 min, 20–30% B in 10 min, 30–90% B in 13 min, 90% B in 2 min.

Proteomic analyses were performed on a Thermo Orbitrap Fusion Lumos mass spectrometer (Thermo Fisher Scientific) using a nano-electrospray ion source with electrospray voltages of 2.2 kV. Xcalibur software in profile spectrum data type was used for data-dependent acquisition. The $MS^1$ full scan was set at a resolution of 60,000 at m/z 200, AGC target 5e4 and maximum IT 50 ms by an Orbitrap mass analyzer (300–1,500 m/z), and that was followed by $MS^2$ scans generated by HCD fragmentation at a resolution of 15,000 at m/z 200, AGC target 5e4 and maximum IT 45 ms. The fixed first mass of the $MS^2$ spectrum, the isolation window, and the normalized collision energy (NCE) were set at 110.0 m/z, 1.6 m/z, and NCE 30%, respectively.

## Mass spectrometry data analysis

Proteome Discoverer 2.1 software (Thermo Fisher Scientific) was used to align the mass spectrometry data with Uniprot *Homo sapiens* (Human). Enzyme specificity was set to trypsin, with a maximum of 2 missed trypsin cleavage sites. Precursor mass tolerance was set to 10 ppm and the fragment ion mass tolerance to 0.02 Da. Also, carbamidomethylation of cysteine was the fixed modification, while oxidation of methionine and protein N-terminal acetylation were variable modifications. Percolator algorithm was used to calculate a 1% false discovery rate at the peptide and protein levels.

## Gene silencing by siRNA

The siRNAs used in this study (Suzhou GenePharma Co., Ltd, Suzhou, China) consisted of the following sequences: ODF2, 5'-AUCUUUAUGUCGCUGAATT-3'; CCDC68, 5'-CUGCGUGAGUCUUAUUUAU-3'; CCDC120, 5'-GGGAGUGGCUAGUCAUGAU-3'; γ-taxilin #1, 5'-GAAGCAACUGCACAUUUCCA GAUUA-3'; γ-taxilin #2, 5'-GGCAAGAAGCAAGCUAGAAUCUCUU-3' (*Makiyama et al., 2018*); α-taxilin #1, 5'- GCCUGAACCAACUCCAGUA-3'; α-taxilin #2, 5'-GCGAGGAGCAUAUCGACAA-3'; and α-taxilin #3, 5'-GCGAGGUAUUCACCACAUU-3'. The negative control was set as 5'- UUCUCCGAAC-GUGUCACGU-3'. All siRNAs were transfected into RPE-1 or HeLa cells by using Lipofectamine 3,000

(Thermo Fisher Scientific) according to the manufacturer's instructions. The cells were analyzed 72 hr post-transfection.

## Establishment of α-taxilin and γ-taxilin knockout cell lines

The CRISPR-Cas9 approach was used to generate α- and γ-taxilin KO cell lines (*Ran et al., 2013*). Each protein's 20-nt guide sequence was designed using the online CRISPR Design Tool at http://www.e-crisp.org/E-CRISP/. α-Taxilin was designed as 5'-GCAGTAGAAGCAGAAGGTCC-3' and γ-taxilin was designed as 5'-CTGTTCGAGCTTCTCCCCCA-3'. The oligo pairs encoding the 20-nt guide sequences were annealed and ligated into a U6-sgRNA plasmid. The U6-sgRNA plasmid, SpCas9-pcDNA3.1, and pcDNA3.1(+)-PuroR were then transfected at a concentration ratio of 2:1:0.5 into HeLa or RPE-1 cells, and puromycin was used to screen or select transfected cells. Clonal cell lines were isolated using Flow cytometry (MoFlo Astrios EQ, Beckman Coulter, Indianapolis, IN, USA) and mutations were detected by subsequent sequencing and immunoblotting.

## Electron microscopy

RPE-1 cells grown on ACLAR 33 C film were fixed by adding double-strength fixative (i.e. 1× fixative: 2.5% glutaraldehyde in 0.1 M PB buffer [pH 7.4]) with an equal volume of culture medium for 3 min at room temperature (RT). After removing the mixed fixative, cells were fixed with 1× fixative for 1 hr at RT, then held overnight at 4 °C. Then, the cells were washed four times (8 min each) with 0.1 M PB buffer, then post-fixed in 1% $OsO_4$ and 0.8% $K_4Fe(CN)_6$ for 1 hr at RT in the dark. After rinsing four times (8 min each) with distilled water, the cells were stained in 1% aqueous uranyl acetate overnight at 4 °C. Following several washes with distilled water, the cells were dehydrated by first using a graded alcohol series (30%, 50%, 70%, 85%, 95%, and 100%, 5 min for each) and then 100% acetone (twice at 5 min each). Subsequently, the cells were infiltrated, embedded in EMbed 812 resin (Electron Microscopy Sciences, Hatfield, PA, USA), and polymerized at 65 °C for 24 hr. After removing the ACLAR 33C film, the resin blocks were trimmed and sectioned using an EM UC7 ultramicrotome (Leica Microsystem, Wetzlar, Germany) with an ultra 35° diamond knife (Diatome Ltd, Nidau, Switzerland). Single-slot copper grids were used to collect 70-nm-thick serial sections that were double stained with uranyl acetate and lead citrate. The grids were then inspected using a Tecnai $G^2$ Spirit BioTWIN transmission electron microscope (Thermo Fisher Scientific) at 120 kV, and images were captured with an attached Orius 832 CCD camera (Gatan, Inc, Pleasanton, CA, USA).

## Immunofluorescence

Cells grown on 18 × 18 mm coverslips were fixed and permeabilized in pre-chilled methanol for 10 min at –20 °C, and then washed three times (10 min each time) in PBS. The cells were then blocked with 4% BSA for 30 min at RT, and then incubated with the first antibody (diluted in 4% BSA) at 4 °C overnight. The slides were then washed three times in PBS buffer for 10 min each time, blocked with 4% BSA for 30 min at RT, and then incubated with the second antibody (Alexa Fluor [Invitrogen, Carlsbad, CA, USA]) for 1 hr at RT. Finally, the cells were stained with DAPI.

Confocal and STED images were acquired using either a TCS SP8 STED 3 X microscopy system with a 100 × 1.4 NA APO oil objective lens and LAS X v.2.0 software (Leica Microsystem), or a STEDYCON microscope platform (Abberior Instruments, Göttingen, Germany). Huygens v.14.10 software (Scientific Volume Imaging, Hilversum, Netherlands) was used for STED nanoscopy image deconvolution. An N-SIM Microscope system equipped with a 100 × 1.49 NA APO oil objective lens (Nikon, Tokyo, Japan) was used to acquire 3D-SIM images. NIS-Elements AR v.4.51 software (Nikon) was used both to acquire the images and for three-dimensional reconstruction. Images were then processed in Photoshop (Adobe, San Jose, CA, USA).

## Immunoprecipitation and immunoblots

For immunoprecipitation, HEK-293T or RPE-1 cells were lysed in immunoprecipitation lysis buffer (50 mM HEPES, 250 mM NaCl, 0.1% Nonidet P-40, 1 mM DTT, 1 mM EDTA, and 10% glycerol, pH = 7.4) on ice for 30 min and then the lysates were centrifuged at 12,000 g for 15 min at 4 °C. Then, either Protein A-Sepharose beads or Protein G Sepharose beads (ab193259 and ab193256, respectively, Abcam, Cambridge, United Kingdom Amersham Biosciences) were added to the supernatants, which were then individually incubated with the appropriate antibodies overnight at 4 °C. The beads

were then washed with immunoprecipitation lysis buffer and collected in SDS loading buffer (50 mM Tris-HCl, pH = 7.4, 2% SDS, 100 mM DTT, 0.025% bromo blue, 10% glycerol), which was then boiled at 100 °C for 10 min to obtain the immunoprecipitation samples.

For immunoblots, SDS-PAGE was used to separate protein samples, which were then transferred onto polyvinylidene difluoride membranes (Sigma, Burlington, MA, USA). The membranes were blocked with 5% non-fat milk for 30 min, and then incubated with primary antibodies either overnight at 4 °C or for 2 hr at room temperature. After incubation, the membranes were then incubated with peroxidase-AffiniPure goat anti-rabbit or goat anti-mouse IgG (H + L) secondary antibodies (1:5000) (#111-035-003 and #115-035-003, respectively, Jackson ImmunoResearch, WestGrove, PA, USA) for 1 hr at RT.

### GST pull-down assay

For the GST pull-down assay, bacterial-expressed GST and GST-α-Taxilin-M (expressed in *Escherichia coli* strain BL21) were bound to glutathione-Sepharose 4B beads (Cytiva) and incubated with bacterial-expressed MBP-γ-Taxilin-M or HEK-293T cell expressed and purified 3×FLAG-CEP170 at 4 °C overnight. After reaction, the complexes were washed at least five times with GST-binding buffer (20 mM HEPES, pH 7.5, 75 mM KCl, 0.1 mM EDTA, 2.5 mM MgCl2, 1 mM DTT, 0.05% Triton X-100), eluted by boiling in SDS-PAGE loading buffer, and subjected to immunoblot with the indicated antibodies.

### Microtubule regrowth assay

RPE-1 cells were grown on 18 × 18 mm coverslips and then the coverslips were embedded on ice for 30 min to depolymerize the cytoplasmic microtubules. The cells were then brought back to 37 °C to allow the microtubules to reform. Cells at 0, 5, and 10 min after rewarming were fixed in 4% para-formaldehyde (pre-warmed to 37 °C) for 10 min at 37 °C. The cells were immunostained with α- and γ-tubulin antibodies.

### Spindle orientation analysis

Spindle orientation was measured from the centrosome pairs' x, y, and z coordinates relative to the slide. Spindle angles measurements were derived from the fixed and immunostained HeLa cells, which were immunostained with an anti-γ-tubulin antibody for spindle poles (red) and an anti-α-tubulin antibody for mitotic spindles (green). The z-stacks and the spindle pole coordinates were measured usinge LAS X v.2.0 software (Leica Microsystem). The spindle angles were then measured using 3D coordinate geometry.

### Measurements and statistical analysis

Immunofluorescence intensities and the diameters of the ring-like SDA protein structures were measured using Image J software v.1.48 (NIH) (*Schneider et al., 2012*). The statistical significances among different groups were determined by two-tailed Student's t-tests or one-way ANOVAs. The data were graphed in Prism 5 (GraphPad, San Diego, CA, USA).

## Acknowledgements

We thank the Flow Cytometry Core at the National Center for Protein Sciences at Peking University in Beijing, China, particularly Liying Du, Jia Luo, Huan Yang, and Hongxia Lv, for their technical help. We also thank the Optical Imaging Core Facility at the National Center for Protein Sciences, particularly Chunyan Shan and Ye Liang for assistance with the 3D-SIM, confocal, and STED microscopy imaging. We thank Yun-Chao Xie and Peng-Yuan Dong for their professional technical assistance in EM sample preparation at the Core Facilities of School of Life Sciences, Peking University. We thank National Center for Protein Sciences at Peking University in Beijing, China, for assistance with protein MS analysis work and we would be grateful to Dong Liu for her help with protein digestion and MS analysis work. We thank all members of the laboratory for their helpful discussions. This work was supported by the National Natural Science Foundation of China (31630092, 31830110, and 31801133) and the National Key Research and Development Program of China, Stem Cell and Translational Research (2016YFA0100501).

## Additional information

### Funding

| Funder | Grant reference number | Author |
|---|---|---|
| National Natural Science Foundation of China | 31630096 | Jianguo Chen |
| National Natural Science Foundation of China | 31830110 | Jianguo Chen |
| National Natural Science Foundation of China | 31801133 | Dandan Ma |
| National Key Research and Development Program of China Stem Cell and Translational Research | 2016YFA0100501 | Jianguo Chen |

The funders had no role in study design, data collection and interpretation, or the decision to submit the work for publication.

### Author contributions

Dandan Ma, Investigation, Writing – original draft; Fulin Wang, Formal analysis, Software; Rongyi Wang, Data curation; Yingchun Hu, Yonglu Tian, Methodology; Zhiquan Chen, Conceptualization; Ning Huang, Writing - review and editing; Yuqing Xia, Methodology, Software; Junlin Teng, Supervision, Writing - review and editing; Jianguo Chen, Project administration, Resources, Supervision

### Author ORCIDs

Fulin Wang ⬥ http://orcid.org/0000-0002-9354-3172
Junlin Teng ⬥ http://orcid.org/0000-0002-7611-0631
Jianguo Chen ⬥ http://orcid.org/0000-0002-8406-011X

### Ethics

All experiments are applied in mammalian cell lines. Neither animals nor primary culture cells are used.

### Decision letter and Author response

Decision letter https://doi.org/10.7554/eLife.73252.sa1
Author response https://doi.org/10.7554/eLife.73252.sa2

## Additional files

### Supplementary files

• Transparent reporting form

### Data availability

All data generated or analysed during this study are included in the manuscript and supporting file. Source data files have been provided for Figures 1–7.

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
