## [Editor Report]

The subdistal appendage (SDA) is a distinct structure of the mother centriole that functions in anchoring microtubules, orientation of spindles and formation of the basal foot of cilia. Though many components in this structure have been identified, two new proteins, i.e. α- and γ-taxilins are identified in this work. By using super-resolution microscopy, biochemical and genetic tools, the precise localization and function of these new proteins have been defined, which would deepen our understanding of this unique structure.

---

## [Decision Letter]

**Decision letter after peer review:**

Thank you for submitting your article "α-/γ-Taxilin are required for centriolar subdistal appendage assembly and microtubule organization" for consideration by *eLife*. Your article has been reviewed by 3 peer reviewers, including Junmin Pan as the Reviewing Editor and Reviewer #1, and the evaluation has been overseen by Piali Sengupta as the Senior Editor.

Based on the reviewer's comments and follow-up discussions, we have made the following suggestions to improve the manuscript. We hope the authors can address these questions and encourage the authors to send a revised submission. Specific reviews are also attached at the end of this letter.

Essential revisions:

(1) APEX2-based proximity mapping of CCDC68 and CCDC120 is not covered in the manuscript with sufficient detail (results, methods, discussion). Since this approach led to identification of taxilins as SDA components in this manuscript and could be used as a resource in future studies, a more comprehensive analysis of the datasets and their validation is required. (1) IF data for representing localization of V5-APEX fusions of CCDC68 and CCDC120 to the SDAs and localized biotinylation at these structures, (2) Western blotting data confirming induced biotinylation, (3) methodology used to analyze the mass spectrometry data, (4) presentation of high confidence proximity interactors of CCDC68 and CCDC120 and their comparative analysis, (5) validation of the datasets- they explained this by identification of γ-tubulin and STIL, how about identification of other SDA components, are they enriched in the datasets?

(2) Α- and Γ-taxilin antibody specificity for immunofluorescence experiments should be confirmed with cells depleted with specific siRNAs as well as with CRISPR KO lines.

(3) Given that authors generated α- and γ-taxilin mutants that only localize to SDAs or proximal ends of centrioles, they have great tools for asking the specific functions of these pools in taxilin-associated phenotypes. In addition to rescue experiments with full length taxilins, rescue experiments with these mutants for microtubule anchoring/nucleation and spindle positioning experiments would provide insight into the mechanisms by which taxilins mediate these functions.

(4) A TEM data of the SDA after loss of taxilin would be nice to show. Or IP of the taxilin-less SDA followed by mass-spectometry to show which proteins are missing from the SDA complex.

*Reviewer #1 (Recommendations for the authors):*

This work aims to study the subdistal appendage (SA) of the mother centriole. Using affinity purification of known SA proteins, the authors identified two new proteins at the subdistal appendages: α and γ taxilins. They worked out the precise localization of these two proteins relative to other components of SA by using super resolution microscopy and determined how they are assembled into the right spot within SA. Furthermore, they examined their functions by using KO cells and demonstrated that loss of these proteins affected MT nucleation and spindle orientation. Overall I think the paper is clearly written and the results are convincing. This work deepens our understanding of the subdistal appendages.

1. Providing some detailed methods:

(1) For the APEX-2 affinity purification, the vendor for the Streptavidin-coated beads and a full name for BP should be provided. The final concentrations of BP and H2O2 should also be provided.

(2) The authors examined spindle angle. However, I could not find how these angles were measured. Please provide detailed methods.

(2) It has been reported that depletion of γ-taxilin induces dissolution of the intercentriolar linkage (Makiyama et al., 2018 Exp Cell Res). I am wondering whether the authors found the same phenomenon. From the figures, I could not see this phenotype. The authors should discuss this point.

*Reviewer #2 (Recommendations for the authors):*

In this manuscript, the authors describe a novel pathway of SDA formation from ODF2 to CEP170 via g-Taxillin and a-Taxillin. These Taxillin proteins associate with each other through coiled-coiled domains and are critical for recruiting CEP170 but not Ninein. Importantly, their deficiency is shown to be defective in anchoring MTs, which is very well consistent with their specific localization to SDAs.

Strengths:

The authors have used powerful imaging techniques to analyze the function of g-Taxillin and a -Taxillin and the data presented in the manuscript were well organized and clear-cut to understand.

Weaknesses:

However, all the data presented in the manuscript are focused on the immortalized cell lines such as RPE-1 and Hela cells. For example, it is important to reveal the expression pattern of Taxillin genes and their in vivo role in animals. At least, their functional roles should be analyzed in primary cells isolated from appropriate tissues. In addition, it will be great to show the overall morphology of SDAs using TEM to see whether Taxillin ablation results in decreasing the size of each SDA. Based on these major issues, it is very difficult to assess the biological significance of the claims.

1. Does CEP170 ablation or knock-down recapitulate the phenotype of Taxillin knock-out in MT anchoring?

2. The authors claim that a-Taxillin does not interact with Ninein in Figure 4A. However, there is a faint band corresponding to endogenous Ninein (lane 3).

3. The data on EB-1 staining presented in Figure 7F is not sufficiently convincing. Better data would have helped to substantiate the authors' conclusions.

4. What happens to Hela or RPE-1 cells upon Taxillin ablation? Authors claim that Taxillins are critical for MT anchoring and spindle orientation. So what? How does they affect the cell behavior?

*Reviewer #3 (Recommendations for the authors):*

In this manuscript, Ma et al. identified α- and γ-taxilin as new subdistal appendage components, reported their role in SDA assembly, organization and examined functional consequences of their cellular loss. They generated proximity maps for known SDA components CCDC68 and CCDC120 using APEX2-based labeling and chose α- and γ-taxilin for further investigation. Using superresolution microscopy, they defined their precise localization at the centriole relative to known SDA components and centriole proximal marker C-Nap1. They then performed depletion or deletion experiments to determine their position in the SDA assembly pathway and found that α- and γ-taxilin localization at the SDA depends on ODF2 and that they are required for centriolar recruitment of CEP170. Finally, they performed loss-of-function experiments and demonstrated that α- and γ-taxilins are required for microtubule anchoring in interphase and mitotic spindle positioning. These findings contribute to our understanding of SDA assembly, organization and function by throughly characterizing two new SDA components. However, these findings do not sufficiently advance our mechanistic understanding of cellular functions and mechanisms of SDAs.

One major weakness of the manuscript is the lack of mechanistic understanding on how α- and γ-taxilin mediate their functions during microtubule organization and spindle positioning. In particular, the presented data do not distinguish whether these functions are specific to the proximal and/or SDA pools of these proteins. Another weakness of the manuscript is the presentation, analysis and discussion of the APEX2-based proximity labeling data. High confidence proximity interactors of taxilins and their comparative analysis with each other and published datasets have to be presented so the resulting datasets provide a resource for the field in the future.

Data in manuscript are high quality, rigorous and well-presented. The paper is well-written with a good logical flow. In particular, the authors used multiple approaches to characterize α- and γ-taxilin as SDA components and to define their precise spatial localization and relationship to other SDA proteins. Despite these attributes, however, I am not convinced the results represent sufficient conceptual advance on our understanding of SDA biogenesis and functions. Loss-of-function experiments showed that α- and γ-taxilin are required for microtubule anchoring and spindle positioning, which were phenotypes previously reported for a number of other SDA proteins. The mechanisms that underlie these functions remain elusive. For example, do proximal or SDA pools of these proteins mediate these functions? How do taxilins regulate astral microtubule length? It might be due to potential functions in microtubule nucleation/polymerization, which have not been investigated.

1 – APEX2-based proximity mapping of CCDC68 and CCDC120 is not covered in the manuscript with sufficient detail (results, methods, discussion). Since this approach led to identification of taxilins as SDA components in this manuscript and could be used as a resource in future studies, a more comprehensive analysis of the datasets and their validation is required. (1) IF data for representing localization of V5-APEX fusions of CCDC68 and CCDC120 to the SDAs and localized biotinylation at these structures, (2) Western blotting data confirming induced biotinylation, (3) methodology used to analyze the mass spectrometry data, (4) presentation of high confidence proximity interactors of CCDC68 and CCDC120 and their comparative analysis, (5) validation of the datasets- They explained this by identification of γ-tubulin and STIL, how about identification of other SDA components, are they enriched in the datasets?

2 – Α- and Γ-taxilin antibody specificity for immunofluorescence experiments should be confirmed with cells depleted with specific siRNAs as well as with CRISPR KO lines.

3 – How to they distinguish between defects in microtubule nucleation and anchoring? Rescue experiments for microtubule nucleation experiments with both mutants.

4 – Given that authors generated α- and γ-taxilin mutants that only localize to SDAs or proximal ends of centrioles, they have great tools for asking the specific functions of these pools in taxilin-associated phenotypes. In addition to rescue experiments with full length taxilins, rescue experiments with these mutants for microtubule anchoring/nucleation and spindle positioning experiments would provide insight into the mechanisms by which taxilins mediate these functions.

5 – The authors show that α-taxilin and γ-taxilin interact. Do they cooperate during SDA biogenesis, microtuble anchoring and/or spindle positioning or do they function independently? For example, can they restore each other's function in loss-of-function experiments.

6 – In addition to spindle mispositoning defects, does depletion of α- or γ-taxilin cause other mitotic phenotypes (i.e. mitotic duration, chromosome segregation etc..).

---

## [Author Response]

Reviewer #1 (Recommendations for the authors):This work aims to study the subdistal appendage (SA) of the mother centriole. Using affinity purification of known SA proteins, the authors identified two new proteins at the subdistal appendages: α and γ taxilins. They worked out the precise localization of these two proteins relative to other components of SA by using super resolution microscopy and determined how they are assembled into the right spot within SA. Furthermore, they examined their functions by using KO cells and demonstrated that loss of these proteins affected MT nucleation and spindle orientation. Overall I think the paper is clearly written and the results are convincing. This work deepens our understanding of the subdistal appendages.1. Providing some detailed methods:(1) For the APEX-2 affinity purification, the vendor for the Streptavidin-coated beads and a full name for BP should be provided. The final concentrations of BP and H2O2 should also be provided.

Thank you for your suggestions. We added detailed information as follows:

1. In the “Results” (page 4), we added, “APEX2-mediated proximal labeling is an approach that uses hydrogen peroxide (H2O2) as an oxidant to catalyze biotin-phenol (BP)”.

2. In the “Materials and methods” section (Pages 23-24), we added the following detail:

“Following treatment with BP (final concentration of 50 µM) for 30 min and H2O2 (final concentration of 1 mM) for 1 min, the biotinylated proteins were enriched by Streptavidin-coated beads (Streptavidin Sepharose High Performace**,** Cytiva, Marlborough, MA, USA**)** for mass spectrographic (MS) analysis.”

(2) The authors examined spindle angle. However, I could not find how these angles were measured. Please provide detailed methods.

Thank you for your suggestion. We added a Spindle orientation analysis subsection in “Materials and methods”, Pages 29-30:

“Spindle orientation analysis

Spindle orientation was measured from the centrosome pairs’ x, y, and z coordinates relative to the slide. Spindle angle measurements were derived from the fixed and immunostained HeLa cells, which were immunostained with an anti-γ-tubulin antibody for spindle poles (red) and an anti-α-tubulin antibody for mitotic spindles (green). The z-stacks and spindle pole coordinates were measured using LAS X v. 2.0 software (Leica Microsystems). The spindle angles were then measured using 3D coordinate geometry.”

2. It has been reported that depletion of γ-taxilin induces dissolution of the intercentriolar linkage (Makiyama et al., 2018 Exp Cell Res). I am wondering whether the authors found the same phenomenon. From the figures, I could not see this phenotype. The authors should discuss this point.

Thank you for your inquiry. In our study, we found a phenotype similar to that of Makiyama *et al.* (2018) and we show it in *Figure 4**—**figure supplement 1C-D* and *Figure 7**—**figure supplement 1C-D*. Also, we discussed that finding on page 17 in the Discussion:

“Our study mainly investigated the manner of assembly and functions of α-taxilin and γ-taxilin at the SDA, but the specific localization of those proteins, as revealed by super resolution microscopy, also showed proximal signals (*Figure 1A-F*) that resembled ninein group patterns (*Mazo et al., 2016*). Previously, γ-taxilin was shown, by interacting with mitosis A-related kinase 2A (Nek2A), to temporally mediate centrosome disjunction before the onset of mitosis (*Makiyama et al., 2018*). This is consistent with our finding that γ-taxilin knockout also resulted in centrosomal disjunction in RPE-1 and HeLa cells (*Figure 4**—**figure supplement 1C-D*; *Figure 7**—**figure supplement 1C-D*). Similarly, α-taxilin depletion in RPE-1 and HeLa cells also resulted in centrosomal disjunction (*Figure 4**—**figure supplement 1E-F*; *Figure 7**—**figure supplement 1E-F*) that resembled the γ-taxilin phenotype. Therefore, the proximal presence of α-taxilin and γ-taxilin at centrioles may work as centrosome linkers regulating centrosome cohesion, and their proximal end assembly rules need further study.”

References

Makiyama T, Higashi S, Sakane H, Nogami S, Shirataki H. 2018. γ-Taxilin temporally regulates centrosome disjunction in a Nek2A-dependent manner. *Exp Cell Res* 362(2): 412-423. doi: 10.1016/j.yexcr.2017.12.004, PMID: 29225051

Mazo G, Soplop N, Wang WJ, Uryu K, Tsou MF. 2016. Spatial control of primary ciliogenesis by subdistal appendages alters sensation-associated properties of cilia. *Dev Cell* 39(4): 424-437. doi: 10.1016/j.devcel.2016.10.006, PMID: 27818179

Reviewer #2 (Recommendations for the authors):[…]1. Does CEP170 ablation or knock-down recapitulate the phenotype of Taxillin knock-out in MT anchoring?

Thank you for the question. *Stefan* (*2009*) demonstrated that siRNA-induced depletion of CEP170 inhibited the formation of a microtubule network. That recapitulates the phenotype of taxillin knock-out in MT anchoring.

2. The authors claim that a-Taxillin does not interact with Ninein in Figure 4A. However, there is a faint band corresponding to endogenous Ninein (lane 3).

Thank you for your observation. Indeed, after a long exposure time, endogenous ninein left a very faint band that was much weaker than those of γ-taxilin and CEP170 (*Figure 4A*). Most likely, an SDA protein complex including α-taxillin, γ-taxilin, CEP170 and ninein was pulled down by the α-taxillin antibody, and the interaction between α-taxillin and ninein was indirect.

3. The data on EB-1 staining presented in Figure 7F is not sufficiently convincing. Better data would have helped to substantiate the authors' conclusions.

Thank you very much for your valuable suggestion. We repeated the experiment and revised *Figure 7F* with those results.

4. What happens to Hela or RPE-1 cells upon Taxillin ablation? Authors claim that Taxillins are critical for MT anchoring and spindle orientation. So what? How does they affect the cell behavior?

Thank you for your important questions. To examine cell behavior, we performed live imaging of WT, as well as *α-taxilin* and γ-taxilin knockout (KO) HeLa cells that were stably expressing mNeonGreen-H2B via Spin Disc (Ultra View Vox, PerkinElmer). The mitotic processes of both taxilins’ KO cells were similar to those in the WT cells, suggesting that neither α-taxilin nor γ-taxilin knockout had any effect on mitotic duration or chromosomal segregation (Author response image 1). However, spindle misorganization has been extensively correlated with cell differentiation, cancer and neurological diseases such as microcephaly and lissencephaly (*Noatynska et al., 2012*), as well as with tubular organ diseases (*Zhong and Zhou, 2017*). Further investigation is needed to determine whether α- and γ-taxilin are involved in disease development.

**Author response image 1. sa2fig1:** A. Effects of taxilin loss on mitotic times. WT HeLa cells and *α-taxilin* or *γ-taxilin* knockout (KO) HeLa cells stably expressing mNeonGreen-H2B were photographed every 3 min for 24 h. Representative still frames from time-lapse experiments are shown with elapsed times shown in each frame. Scale bar, 5 µm. B. Quantification of the time intervals from nuclear envelope breakdown to anaphase onset. Data were mean ± SEM of three independent experiments. n>100. One-way ANOVA was used for statistical analysis. n.s., not significant.

References

Noatynska A, Gotta M, Meraldi P. 2012. Mitotic spindle (DIS) orientation and DISease: cause or consequence? *J Cell Biol* 199(7): 1025-1035. doi: 10.1083/jcb.201209015., PMCID: PMC3529530

Stefan L. 2009. Functional characterisation of the centrosomal protein Cep170. Dissertation, LMU München: Faculty of Biology.

Zhong T, Zhou J. Orientation of the Mitotic Spindle in the Development of Tubular Organs. 2017. *Cell Biochem.* 2017 118(7): 1630-1633. doi: 10.1002/jcb.25865, PMID: 28059469

Reviewer #3 (Recommendations for the authors):[…]1 – APEX2-based proximity mapping of CCDC68 and CCDC120 is not covered in the manuscript with sufficient detail (results, methods, discussion). Since this approach led to identification of taxilins as SDA components in this manuscript and could be used as a resource in future studies, a more comprehensive analysis of the datasets and their validation is required. (1) IF data for representing localization of V5-APEX fusions of CCDC68 and CCDC120 to the SDAs and localized biotinylation at these structures, (2) Western blotting data confirming induced biotinylation, (3) methodology used to analyze the mass spectrometry data, (4) presentation of high confidence proximity interactors of CCDC68 and CCDC120 and their comparative analysis, (5) validation of the datasets- They explained this by identification of γ-tubulin and STIL, how about identification of other SDA components, are they enriched in the datasets?

Thank you for the valuable suggestion. We added several details in the results, methods and discussion part as follows:

1. IF data showing localization of the V5-APEX fusions of CCDC68 or CCDC120 to the SDAs and the biotinylated proteins at those areas (revised *Figure 1—figure supplement 1A-B*).

2. Western blotting data confirming endogenous biotinylated proteins mediated by CCDC68 or CCDC120 proximal labeling were added (revised *Figure 1—figure supplement 1C-D*).

3. We added detailed methodology that was used to analyze the mass spectrometry data (Page 25):

“Proteome Discoverer 2.1 software (Thermo Fisher Scientific) was used to align the mass spectrometry data with Uniprot *Homo sapiens* (Human). Enzyme specificity was set to trypsin, with a maximum of 2 missed trypsin cleavage sites. Precursor mass tolerance was set to 10 ppm and the fragment ion mass tolerance to 0.02 Da. Also, carbamidomethylation of cysteine was the fixed modification, while oxidation of methionine and protein N-terminal acetylation were variable modifications. Percolator algorithm was used to calculate a 1% false discovery rate at the peptide and protein levels.”

4. We have presented CCDC68 and CCDC120 proximity interactors and their comparative analyses. Please see “Figure 1—figure supplement 1—source data 2. CCDC68 and CCDC120 proximity spectrometry data” in revised source data.

5. In the MS datasets, we identified a series of centrosomal proteins, including γ-tubulin, CC2D1A, CCDC78, CCDC138, etc*.* α-Taxilin and γ-taxilin, which were localized at the SDA, were also included. However, other already known SDA proteins, such as ODF2, ninein, and CEP170 were not included in the datasets because of their low protein levels in the cells (for ODF2, see *Figure 3A-B*) or their relatively longer distance from CCDC68 and CCDC120 (for ninein and CEP170, *Figure 2*).

2 – α- and -taxilin antibody specificity for immunofluorescence experiments should be confirmed with cells depleted with specific siRNAs as well as with CRISPR KO lines.

Thank you for the valuable suggestion. To validate α-taxilin and γ-taxilin antibody specificities, we added immunofluorescence experiments using cells with specific siRNAs as well as CRISPR KO lines. Results summary:

1. The antibody-stained α-taxilin and γ-taxilin were prominently focused at the centrosome, as demonstrated by their co-localizations with γ-tubulin in human RPE-1 cells. The fluorescence intensity of α-taxilin and γ-taxilin at the centrosome decreased after siRNA knockdown (*Figure 1—figure supplement 2D*). Those results indicated both the specificities of antibodies and the efficiency of siRNAs.

2. In the *α-taxilin* and *γ-taxilin* knockout RPE-1 and HeLa cells, α-taxilin and γ-taxilin fluorescence signals at the centrosome disappeared (*Figure 4—figure supplement 1C and E*; *Figure 7—figure supplement 1C and E* in the revised manuscript).

3 – How to they distinguish between defects in microtubule nucleation and anchoring? Rescue experiments for microtubule nucleation experiments with both mutants.

Thank you for the consideration. The SDA is well known for its roles in microtubule anchoring (*Chong et al., 2020*). During the microtubule regrowth assay, the microtubules nucleate at the centrosome first (in less than 5 minutes), and then some of them anchor at the SDA (after 5 minutes). We focused mainly on the microtubule regrowth process during the 5 to 10 minutes, during which time the microtubules became anchored at the centrosome.

Our result showed that γ-tubulin fluorescence intensity at the interphase centrosome remained unchanged following both α-taxilin and γ-taxilin siRNA-induced depletion, thus suggesting that microtubule nucleation may not be affected (revised *Figure 6—figure supplement 1F-G*).

4 – Given that authors generated α- and γ-taxilin mutants that only localize to SDAs or proximal ends of centrioles, they have great tools for asking the specific functions of these pools in taxilin-associated phenotypes. In addition to rescue experiments with full length taxilins, rescue experiments with these mutants for microtubule anchoring/nucleation and spindle positioning experiments would provide insight into the mechanisms by which taxilins mediate these functions.

Thank you very much for the valuable suggestion. We added rescue experiments with full-length taxilins, as well as with the mutants for microtubule anchoring and spindle orientation (pages 13 and 14). Results summary:

1. The compromised microtubule regrowth, triggered by depletion of α-taxilin or γ-taxilin, could be partly rescued by overexpression of α- or γ-taxilin full-length, but not by the M2 region deletion mutants of either taxilin, which we previously showed to lose their SDA localization. This suggests that the SDA localization of both α-taxilin and γ-taxilin have indispensable roles in controlling microtubule anchoring at the interphase centrosome (*Figure 6B-E* in revised manuscript).

2. The average spindle angle was significantly increased in *α-taxilin* or *γ-taxilin* KO cells. This was rescued by overexpression of either full-length α-taxilin or γ-taxilin, as well as by the α-taxilin M3 deletion mutant or the γ-taxilin M1 deletion mutant, but not by the M2 deletion mutant for each taxilin. These results confirm α- and γ-taxilin roles in maintaining proper spindle orientation during metaphase (*Figure 7D-E* in revised manuscript).

5 – The authors show that α-taxilin and γ-taxilin interact. Do they cooperate during SDA biogenesis, microtuble anchoring and/or spindle positioning or do they function independently? For example, can they restore each other's function in loss-of-function experiments.

Thank you for that thoughtful query. We discussed this from three perspectives:

(1) Since the two taxilins interact with each other in a direct manner via their coiled-coil domains, it is likely that they cooperate during SDA biogenesis (*Figure 4E-I* in the revised manuscript).

(2) TEM images revealed that SDA structural damage was worse in *γ-taxilin* KO RPE-1 cells than in *α-taxilin* KO RPE-1 cells, thus suggesting that γ-taxilin was likely upstream of α-taxilin at the SDA (*Figure 6A* in the revised manuscript).

(3) Since γ-taxilin depletion affected α-taxilin centrosomal localization (*Figure 4D* in the revised manuscript), overexpressed α-taxilin was thought unable to localize at the centrosome, let alone to rescue γ-taxilin-related functions. Conversely, α-taxilin depletion did not affect γ-taxilin centrosomal localization (*Figure 4D* in the revised manuscript), so the overexpressed γ-taxilin was most likely unable to rescue the α-taxilin-related functions.

Overall, we think the taxilins cooperate during SDA biogenesis and their functions are dependent on each other.

6 – In addition to spindle mispositoning defects, does depletion of α- or γ-taxilin cause other mitotic phenotypes (i.e. mitotic duration, chromosome segregation etc..).

Thank you for the question. To assay for cell cycle progression defects (via Spin Disc, Ultra View Vox, PerkinElmer), we performed live imaging of WT HeLa cells and *α-taxilin* and γ-taxilin knockout HeLa cells that stably expressed mNeonGreen-H2B. Both the WT and the taxilin KO cells had similar mitotic times, suggesting that neither α-taxilin nor γ-taxilin knockout had any effect on mitotic duration or chromosome segregation.